# DEEP LIFETIME CLUSTERING

## ABSTRACT

The goal of lifetime clustering is to develop an inductive model that maps subjects into $K$ clusters according to their underlying (unobserved) lifetime distribution. We introduce a neural-network based lifetime clustering model that can find cluster assignments by directly maximizing the divergence between the empirical lifetime distributions of the clusters. Accordingly, we define a novel clustering loss function over the lifetime distributions (of entire clusters) based on a tight upper bound of the two-sample Kuiper test p-value. The resultant model is robust to the modeling issues associated with the unobservability of termination signals, and does not assume proportional hazards. Our results in real and synthetic datasets show significantly better lifetime clusters (as evaluated by C-index, Brier Score, Logrank score and adjusted Rand index) as compared to competing approaches.

## 1 INTRODUCTION

Survival analysis is widely used to model the relationship between subject covariates and the time until a particular *terminal* event of interest (e.g., death, or quitting of social media) that marks the end of all activities (or measurements) of that subject (known as the *lifetime* of the subject). For instance, a subject's logins to a social network are her activities and the time until she quits the social network permanently is her lifetime.

The lifetime of a subject can be unobserved for two possible reasons: (a) the *terminal* event was right-censored, i.e., the subject did not have a terminal event within the finite data-collection period, or (b) the terminal events are inherently unobservable. Right-censoring happens for instance when a patient is still alive at the time of data-collection. Unobservability happens for instance, in social networks, when a subject simply stops using the service but does not provide a clear *termination signal* by deleting her account. In such a scenario, the terminal events remain unobserved for most if not all subjects, even if the subjects quit the service within the data-collection period.

Numerous survival methods have been proposed (Witten and Tibshirani, 2010a; Hothorn et al., 2006; Ishwaran et al., 2008) to predict the lifetime of a subject given her covariates and her activities/measurements for a brief initial period of time, while also accounting for right-censoring. More recent deep learning models for lifetime prediction (Lee et al., 2018; Ren et al., 2018; Chapfuwa et al., 2018) have achieved much success due to their flexibility to model complex relationships, and by avoiding limiting assumptions like parametric lifetime distributions (Ranganath et al., 2016) and proportional hazards (Katzman et al., 2018). In scenarios where terminal events are never observed (unobservability), it is a standard practice to introduce artificial termination signals through a predefined "timeout" for the period of inactivity, i.e., a social network user inactive for $m$ months has her last observed activity declared a terminal event. Such a specification is typically arbitrary and can adversely affect the analysis depending on the "timeout" value used.

Notwithstanding the fact that lifetimes are hard to predict without termination signals in the training data, we are generally interested in clustering subjects based on their underlying lifetime distribution to improve decision-making. Applications include identifying disease subtypes (Gan et al., 2018), understanding the implications of distinct manufacturing processes on machine parts, and qualitatively analyzing different survival groups in a social network. Although accurately predicting time-to-terminal-event for an individual is important in a variety of applications, lifetime clustering plays a complementary role and provides a more holistic picture.

Lifetime clustering remains a relatively unexplored topic despite being an important tool. Although traditional unsupervised clustering methods such as $k$-means and hierarchical clustering are popular

for this task (Bhattacharjee et al., 2001; Sørlie et al., 2001; Bullinger et al., 2004), they may produce clusters that are entirely uncorrelated with lifetimes (Gaynor and Bair, 2013). Semi-supervised clustering (Bair and Tibshirani, 2004) and supervised sparse clustering (Witten and Tibshirani, 2010b) employ a two-stage lifetime clustering process: (i) identify covariates associated with lifetime using Cox scores (Cox, 1992), and (ii) treat these covariates differently while performing $k$-means clustering. They assume proportional hazards (i.e., constant hazard ratios over time) and require the presence of termination signals. Furthermore, a decoupled two-stage process such as the above is not guaranteed to obtain clusters with maximally distinct lifetime distributions; rather, we require an end-to-end learning framework that prescribes a loss function specifically over the lifetime distributions of different clusters.

In this work we tackle the important task of inductive *lifetime clustering* without assuming proportional hazards, while also smoothly handling the unobservability of *termination* signals.

**Contributions. (i)** We introduce *DeepCLife*, an inductive neural-network based lifetime clustering model that finds cluster assignments by maximizing the divergence between empirical (non-parametric) lifetime distributions of different clusters. Whereas the subjects of different clusters have distinct lifetime distributions, within a cluster all subjects share the same lifetime distribution even if they have different lifetimes. Our model is robust to the modeling issues associated with the unobservability of termination signals and does not assume proportional hazards.
**(ii)** We define a novel clustering loss function over empirical lifetime distributions (of entire clusters) based on the Kuiper two-sample test. We provide a tight upper bound of the Kuiper p-value with easy-to-compute gradients facilitating its use as a loss function, which until now was prohibitively expensive due to the test's infinite sum.
**(iii)** Finally, our results on real and synthetic datasets show that the proposed lifetime clustering approach produces significantly better clusters with distinct lifetime distributions (as evaluated by Logrank score, C-index, Brier Score and Rand index) as compared to competing approaches.

## 2 FORMAL FRAMEWORK

In this section, we formally define the statistical framework underlying the clustering approach introduced later in the paper. *Notation remark*: We use superscript $(u)$ to refer to variables indexed by a subject $u$ and use subscript $k$ to refer to random variables indexed by a random subject of cluster $k$. See Table 1 for a complete list of all the variables and their meanings.

We assume that there are distinct underlying clusters with different event processes describing the activity events of a random subject (e.g., logins to a social network, measurements of a patient) in the respective clusters. To make these notions more formal, we introduce our event process.

**Definition 1** (Abstract Event Process). *Consider the $k$-th cluster. The Random Marked Point Process (RMPP) for the activity events is $\Phi_k = \{X_k, \{(A_{k,i}, M_{k,i}, Y_{k,i})\}_{i \in \mathbb{N}}, \Theta_k\}$ over discrete times $t = 0, 1, \ldots$, where $X_k$ is the random variable representing the covariates of a random subject in cluster $k$, $\Theta_k$ is the time to the zeroth activity event (joining), $M_{k,i}$ represent covariates of event $i$, $Y_{k,i}$ is the inter-event time between the $i$-th and the $(i-1)$-st activity events (e.g., logins), and $A_{k,i} = 1$ indicates an event with a termination signal (death), otherwise $A_{k,i} = 0$. All these variables may be arbitrarily dependent. This definition is model-free, i.e., we will not prescribe a model for $\Phi_k$.*

We observe the RMPP over a time window $[0, t_{\mathrm{m}}]$. Using the above formalism, we define the true lifetime of a subject in cluster $k$ as the sum of all inter-event times until termination signal $A_{k,i} = 1$ is observed.

**Definition 2** (True lifetime). *The random variable that defines the true lifetime until the terminal event of a subject in cluster $k$ is $T_k := \max_i \left( \sum_{i' \leq i} Y_{k,i'} \prod_{i'' < i} (1 - A_{k,i''}) \right).$*

The true lifetime $T_k$ is unobserved if: (a) the terminal event $A_{k,i} = 1$ does not occur within the observation time period $[0, t_{\mathrm{m}}]$, or (b) the termination signals are inherently unobservable. Now, the true lifetime distribution of a subject in cluster $k$ is defined as the probability that the subject has at least one more activity event after time $t$, and is given by $S_k(t) := P[T_k > t] = 1 - F_k(t)$, $t \in \mathbb{N} \cup \{0\}$, where $F_k(t)$ is the underlying cumulative distribution function (CDF) of $T_k$. Note again that all the subjects of a cluster share the same lifetime distribution.

Table 1: Table of notations.

| Notation | Description | Notation | Description |
|---|---|---|---|
| $\square^{(u)}$ | Variables indexed by a subject $u$ | $T$ | True lifetime of the subject (unobserved) |
| $\square_k$ | Random variables indexed by a random subject of cluster $k$ | $H$ | Observed lifetime of the subject |
| $\Theta$ | Joining time of the subject | $\chi$ | Time elapsed since the last observed activity event before $t_{\mathrm{m}}$ |
| $X$ | Covariates of the subject | $F$ | CDF of the true lifetime $T$ |
| $Y_i$ | Inter-event time between $i$-th and $(i-1)$-st activity events | $S$ | True lifetime distribution ($\equiv$ CCDF of true lifetime $T$) |
| $M_i$ | Covariates of event $i$ | $\hat{S}$ | Empirical lifetime distribution |
| $A_i$ | $A_i = 1$ denotes whether $i$-th activity event is terminal | $\hat{S}[t]$ | $\hat{S}$ indexed by $t \in \{0,1,2\ldots\}$ |

**Training data.** Our training data $\mathcal{D}$ consists of subjects from $K$ underlying (hidden) clusters with distinct lifetime distributions $S_k$ for $k \in \{1, \ldots, K\}$. For a subject $u \in \mathcal{D}$, we observe the following quantities $\{X^{(u)}, \{(M_i^{(u)}, Y_i^{(u)})\}_{i=1}^{Q^{(u)}(t_{\mathrm{m}})}, \Theta^{(u)}\}$, where $X^{(u)}, Y_i^{(u)}, M_i^{(u)}$ and $\Theta^{(u)}$ are analogously defined as in Definition 1, but for a given subject $u$. $Q^{(u)}(t)$ is the number of observed activity events of $u$ after her joining and before time $t$. The training data may or may not contain the termination signals, $\{A_i^{(u)}\}_{i=1}^{Q^{(u)}(t_{\mathrm{m}})}$, where $A_i^{(u)} = 1$ indicates that event $i$ was a terminal event for subject $u$ (e.g., death). Termination signals are typically available in healthcare applications, whereas in the case of social networks, we usually do not observe the termination signal (i.e., account deletion) for *any* subject.

We define the *observed* lifetime of a subject $u$ as $H^{(u)} := \sum_{i=1}^{Q^{(u)}(t_{\mathrm{m}})} Y_i^{(u)}$, i.e., the sum of all the inter-event times within the observation period $[0, t_{\mathrm{m}}]$. In the absence of termination signals, we additionally define *inactive period* as the time elapsed since the last observed event of subject $u$, given by $\chi^{(u)} := t_{\mathrm{m}} - \Theta^{(u)} - H^{(u)}$. We use $H^{(u)}$ and $\chi^{(u)}$ only during training. Figure 1 shows the true lifetime, the observed lifetime, and the inactive period for four users of two different clusters in the training data. The events are observed (denoted by solid circles) only till the time of measurement $t_{\mathrm{m}}$, whereas the rest of the events are right-censored (denoted by solid diamonds). The termination signal (denoted by dotted diamonds) may be unobserved even if it occurs before $t_{\mathrm{m}}$ (eg., $u_3$).

**Test data.** For a new test subject, we can only observe her covariates and her activity events for a brief initial period of time $\tau$ since her joining. Formally, we observe the following quantities $\{X^{(u)}, \{(M_i^{(u)}, Y_i^{(u)})\}_{i=1}^{Q^{(u)}(\Theta^{(u)}+\tau)}, \Theta^{(u)}\}$. The goal of inductive lifetime clustering is to be able to assign a cluster for a new unseen subject within $\tau$ time of her joining. Note that since $H^{(u)}$ and $\chi^{(u)}$ are tied to $t_m$, they cannot be computed for test subjects.

Lastly, we formally define our clustering problem.

**Definition 3** (Clustering problem). *Consider a dataset $\mathcal{D}$ with $N$ subjects. Our goal is to find a mapping $\kappa : \left(X^{(u)}, \{(M_i^{(u)}, Y_i^{(u)})\}_{i=1}^{Q^{(u)}(\Theta^{(u)}+\tau)}\right) \to \{1, \ldots, K\}$, that inductively maps subject covariates and observed activity events for a brief initial period of time $\tau$ into clusters, such that the divergence $\Delta$ between the empirical lifetime distributions of these clusters is maximized, i.e.,*

$$\kappa^{\star} = \arg\max_{\kappa \in \mathcal{K}} \min_{\substack{i,j \in \{1\ldots K\}, \\ i \neq j}} \Delta\big(\hat{S}_i(\kappa), \hat{S}_j(\kappa)\big), \tag{1}$$

*where $\mathcal{K}$ is a set of all mappings, $\hat{S}_k(\kappa)$ is the empirical lifetime distribution of subjects in $\mathcal{D}$ mapped to cluster $k$ through $\kappa$, and $\Delta$ is an empirical distribution divergence measure.*

$\kappa^{*}$ optimized in this fashion guarantees that subjects in different clusters have different lifetime distributions [1].

---

[1] See Appendix A.5 for why we wish to maximize the *minimum* divergence across all pairs instead of the *sum* of divergences across all pairs.

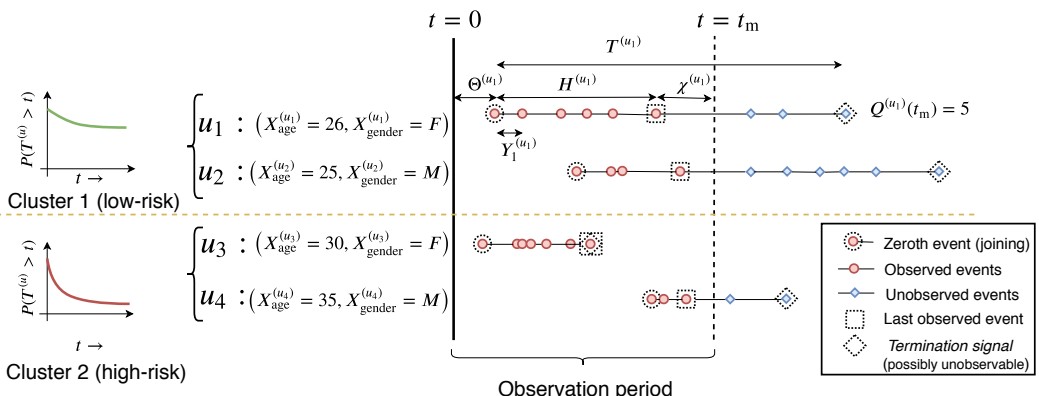

Figure 1: Depicting two clusters (low-risk and high-risk; as shown by the true lifetime distributions) following different RMPPs, each with two subjects. $T^{(u)}$ is the true lifetime of subject $u$ (may be unobserved due to right censoring or due to unobservability of termination signals). $H^{(u)}$ is her *observed* lifetime, the period between the first and the last observed event. $\chi^{(u)}$ is the time between the last observed event and $t_{\mathrm{m}}$, and $Q^{(u)}(t_{\mathrm{m}})$ is the number of events of $u$ after her joining and before $t_{\mathrm{m}}$.

## 3 THE DEEPCLIFE MODEL

In this section, we propose a practical lifetime clustering approach using neural networks that optimizes Equation (1). Let $\mathcal{D}$ be the training data as defined in Section 2. Since we want to maximize divergence between empirical lifetime distributions, we assume discrete times (relative to subject joining), $t \in \{0, 1, \ldots, t_{\max}\}$, where $t_{\max} = \max_{u \in \mathcal{D}} H^{(u)}$ is the maximum observed lifetime of any subject $u \in \mathcal{D}$. Note that it is sufficient to define the empirical distribution till $t_{\max}$, since we have not observed any subject with lifetime greater than $t_{\max}$.

### 3.1 CLUSTER ASSIGNMENTS : $\alpha_k^{(u)}(W_1)$

We define a neural network $g$ that takes user covariates and the event data for a subject $u$ during a brief initial period $\tau$ after her joining as input, and outputs her cluster assignment probabilities, $\alpha_k^{(u)}(W_1)$ for all $k \in \{1, \ldots K\}$,

$$\vec{\alpha}^{(u)}(W_1) = g\left(X^{(u)}, \{M_i^{(u)}, Y_i^{(u)}\}_{i=1}^{Q^{(u)}(\Theta^{(u)}+\tau)}; W_1\right) , \tag{2}$$

where $W_1$ are the weights of the neural network. The final layer of $g$ is a softmax layer with $K$ units. Figure 2a depicts $g$ as a feedforward neural network with $L-1$ hidden layers, although our model is not restricted to a feedforward architecture. In our experiments, we compute summary statistics over the observed events $\{M_i^{(u)}, Y_i^{(u)}\}_i$ in order to make it compatible with the feedforward architecture.

### 3.2 PROBABILITY OF TERMINATION: $\beta^{(u)}(W_2)$

During the training of our model, we require termination signals or a probabilistic estimation of the termination signals in order to write the likelihood of the model. Given the model parameter $W_2$, we define $\beta^{(u)}(W_2)$ as the probability that the last observed event of $u$ was terminal, i.e., $u$ will have no future activity events after the last observed event.

*If the termination signals $A_i^{(u)}$ are observed in the training data $\mathcal{D}$ (e.g., healthcare), clearly $\beta^{(u)}(W_2) := A_{Q^{(u)}(t_m)}^{(u)}$ (i.e., $W_2$ is ignored).*

When such termination signals are unobservable (e.g., social network), existing survival methods commonly use a timeout window of predefined size $W_{\text{fixed}}$ over the inactive period $\chi^{(u)}$ to specify the probability of termination as $\beta^{(u)}(W_{\text{fixed}}) := \mathbf{1}[\chi^{(u)} > W_{\text{fixed}}]$. However, such specification is arbitrary and precludes any learning of the window size parameter $W_{\text{fixed}}$. Instead, we model the latent termination probabilities $\beta^{(u)}(W_2)$ using a smooth non-decreasing function of $\chi^{(u)}$, i.e., higher the period of inactivity, higher the probability that the last observed event was terminal.

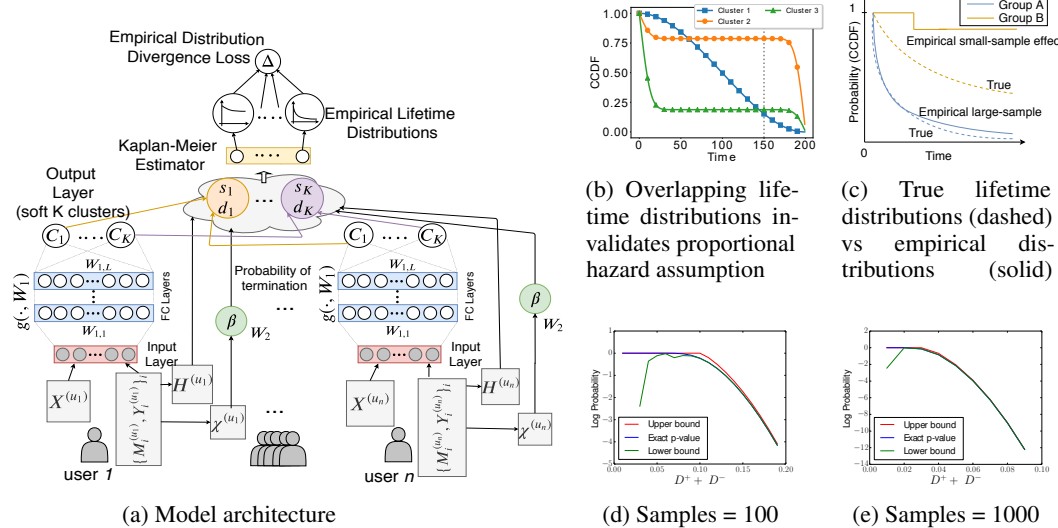

(a) Model architecture

(b) Overlapping lifetime distributions invalidates proportional hazard assumption

(c) True lifetime distributions (dashed) vs empirical distributions (solid)

(d) Samples = 100

(e) Samples = 1000

Figure 2: (a) Feedforward neural network $g(; W_1)$ outputs the cluster assignments for a batch of users. The cluster assignments along with the probability of termination is used to obtain lifetime distributions of each cluster using Kaplan-Meier estimator. Finally, logarithm of Kuiper p-value upper bound is used as the divergence loss $\Delta$. (b) Lifetime distributions can have different shapes and can cross each other, violating proportional hazards assumptions. (c) Divergence metric must account for the uncertainty in the distributions, otherwise divergence maximization leads to imbalanced clusters. (d-e) Upper and lower bounds of the logarithm of Kuiper p-value when varying the Kuiper statistic $D^+ + D^-$.

If the termination signals $A_i^{(u)}$ are unobservable in the training data $\mathcal{D}$ (e.g., social network), we use $\beta^{(u)}(W_2) := 1 - e^{-W_2 \cdot \chi^{(u)}}$ where $W_2 \geq 0$.

### 3.3 Empirical lifetime distribution of cluster $k$ : $\hat{S}_k(W_1, W_2; \mathcal{D})$

Given the training data $\mathcal{D}$ and model parameters $W_1$ and $W_2$, we can obtain the soft cluster assignments $\alpha_k^{(u)}(W_1)$ and the probability of termination $\beta^{(u)}(W_2)$ for all subjects $u \in \mathcal{D}$ and clusters $k \in \{1 \ldots K\}$ as shown in Section 3.1 and Section 3.2. In this subsection, we obtain the empirical lifetime distribution of all the clusters $k = 1 \ldots K$, using the Kaplan-Meier estimates (Kaplan and Meier, 1958). We do not assume a parametric form for the lifetime distribution, and rather use empirical distributions in our optimization to allow lifetime curves of any shape (Figure 2b). Kaplan-Meier estimates are a maximum likelihood estimate of the lifetime distribution of a set of subjects assuming (a) hard memberships (each subject entirely belongs to the set) and (b) the presence of termination signals. We modify the estimates to account for partial memberships and probability of termination instead.

**Proposition 1.** *Given the training data $\mathcal{D}$, a cluster $k$, the cluster assignment probabilities $\{\alpha_k^{(u)}(W_1)\}_{u \in \mathcal{D}}$, and the probabilities of termination $\{\beta^{(u)}(W_2)\}_{u \in \mathcal{D}}$, the maximum likelihood estimate of the empirical lifetime distribution of cluster $k$ is given by,*

$$\hat{S}_k(W_1, W_2; \mathcal{D})[t] = \prod_{j=0}^{t} \frac{s_k(W_1; \mathcal{D})[j] - d_k(W_1, W_2; \mathcal{D})[j]}{s_k(W_1; \mathcal{D})[j]} , \qquad (3)$$

*for all $t \in \{0, 1, \ldots, t_{max}\}$, where, $s_k(W_1; \mathcal{D})[j] = \sum_{u \in \mathcal{D}} \mathbf{1}[H^{(u)} \geq j] \cdot \alpha_k^{(u)}(W_1)$, is the expected number of subjects in cluster $k$ who are at risk (of termination) at time $j$, and, $d_k(W_1, W_2; \mathcal{D})[j] = \sum_{u \in \mathcal{D}} \mathbf{1}[H^{(u)} = j] \cdot \beta^{(u)}(W_2) \cdot \alpha_k^{(u)}(W_1)$, is the expected number of subjects that are predicted to have had a terminal event at time $j$.*

The proof is presented in the supplementary material.

### 3.4   EMPIRICAL DISTRIBUTION DIVERGENCE LOSS : $\Delta(\hat{S}_a, \hat{S}_b)$

We rewrite the objective function of lifetime clustering from Definition 3 with respect to the model parameters $W_1$ and $W_2$ as follows,

$$W_1^*, W_2^* = \arg\max_{W_1, W_2} \min_{\substack{i,j \in \{1...K\}, \\ i \neq j}} \Delta\left(\hat{S}_i, \hat{S}_j\right) ,\tag{4}$$

where $\hat{S}_i$ is the empirical distribution, a shorthand for the vector $\hat{S}_i(W_1, W_2; \mathcal{D})$, and $\Delta$ is a divergence measure between two empirical distributions.

We note the following essential requirements for the divergence measure $\Delta$: (a) $\Delta$ defined over *empirical* distributions must take into account sample sizes, (b) $\Delta$ should have easy-to-compute gradients since it is used as an objective function to train neural networks, and (c) $\Delta$ should not assume proportional hazards, and should allow for crossing lifetime curves (see Figure 2b).

Divergence measures such as Kullback-Leibler (Kullback and Leibler, 1951) and MMD (Gretton et al., 2012) fulfill **(b, c)** but not **(a)**, and will result in sample anomalies as depicted in Figure 2c (also seen in our experiments: Figure 3d). Logrank test (Mantel, 1966), commonly used for comparing lifetime distributions, fulfills **(a, b)**, but has low statistical power when the proportional hazards assumption is not met (Peto and Peto, 1972; Bland and Altman, 2004) (e.g., Figure 2b). Finally, p-value from two-sample tests such as the Kolmogorov-Smirnov (K-S) test (Massey Jr, 1951) fulfill **(a, c)**, but not **(b)** as they require the computation of an infinite sum, resulting in an impractical objective function unless heuristic approximations are made.

We propose to use the Kuiper test (Kuiper, 1960), a two-sample test closely related to the K-S test with increased statistical power in distinguishing distribution tails (Tygert, 2010). Specifically, we define $\Delta(\hat{S}_a, \hat{S}_b) := -\log(KD(\hat{S}_a, \hat{S}_b))$, where KD is the p-value from the Kuiper test between $\hat{S}_a$ and $\hat{S}_b$. Our choice of the Kuiper test is because it is amenable to upper and lower bounds as shown next, thus avoiding the prohibitive heuristic approximations of infinite sums.

**Proposition 2.** *(Bounds for Kuiper p-value.) Given two empirical lifetime distributions $\hat{S}_a$ and $\hat{S}_b$ with discrete support and sample sizes $n_a$ and $n_b$ respectively, define the maximum positive and negative separations between them,*

$$\hat{D}_{a,b}^+ = \sup_{t \in \{0,...t_{max}\}} (\hat{S}_a[t] - \hat{S}_b[t]) , \qquad \hat{D}_{a,b}^- = \sup_{t \in \{0,...t_{max}\}} (\hat{S}_b[t] - \hat{S}_a[t]) .$$

*The Kuiper test p-value Kuiper (1960) gives the probability that $\Lambda$, the empirical deviation for $n_a$ and $n_b$ observations under the null hypothesis $S_a = S_b$ [2], exceeds the observed value $V = \hat{D}_{a,b}^+ + \hat{D}_{a,b}^-$:*

$$KD(\hat{S}_a, \hat{S}_b) \equiv P[\Lambda > V] = 2\sum_{j=1}^{\infty} (4j^2\lambda_{a,b}^2 - 1)e^{-2j^2\lambda_{a,b}^2},\tag{5}$$

*$\lambda_{a,b} = (\sqrt{M_{a,b}} + 0.155 + \frac{0.24}{\sqrt{M_{a,b}}})V$ and $M_{a,b} = \frac{n_a n_b}{n_a + n_b}$ is the effective sample size. Then, the upper bound [3] for the Kuiper p-value is,*

$$KD(\hat{S}_a, \hat{S}_b) \leq \min\Big(1, \ 2 \cdot \mathbf{1}[r_{a,b}^{(lo)} \geq 1] \cdot \big(w(r_{a,b}^{(lo)}, \lambda_{a,b}) - w(1, \lambda_{a,b}) + v(r_{a,b}^{(lo)}, \lambda_{a,b})\big)$$

$$+ v(r_{a,b}^{(up)}, \lambda_{a,b}) - w(r_{a,b}^{(up)}, \lambda_{a,b})\Big),\tag{6}$$

*where $v(r, \lambda) = (4r^2\lambda^2 - 1)e^{-2r^2\lambda^2}, w(r, \lambda) = -re^{-2r^2\lambda^2}, r_{a,b}^{(lo)} = \lfloor \frac{1}{\sqrt{2}\lambda_{a,b}} \rfloor$, and, $r_{a,b}^{(up)} = \lceil \frac{1}{\sqrt{2}\lambda_{a,b}} \rceil$.*

The proof is presented in the supplementary material.

Figures 2d and 2e show empirical results of the upper and lower bounds as a function of $V = D^+ + D^-$ against the expensive heuristic computation where the infinite sum in Equation (5) is approximated with $10^4$ terms (Press et al., 1996). We optimize Equation (4) with $\Delta(\hat{S}_a, \hat{S}_b) := -\log(KD^{(up)}(\hat{S}_a, \hat{S}_b))$ where $KD^{(up)}$ denotes the Kuiper p-value upper bound in Equation (6). $\Delta$ defined this way satisfies all three requirements we described in the beginning of Section 3.4: takes sample sizes into account, has closed form expression with easy-to-compute gradients, and does not assume proportional hazards.

---

[2]The test is typically defined using CDFs but it is equivalent to use lifetime distributions (CCDFs) instead.

[3]Lower bound is presented in the supplementary material.

**Implementation.** We implement our clustering procedure as a feedforward neural network in Pytorch (Paszke et al., 2017) and use ADAM (Kingma and Ba, 2014) to optimize Equation (4). Each iteration of the optimization takes as input a batch of subjects, generates a single value for the set loss, calculates the gradients, and updates the parameters $W_1$ and $W_2$. We show in the supplementary material that the time and space complexity per iteration of the proposed approach are $O(KB + K^2 t_{max})$ and $O(K t_{max})$ respectively, where $K$ is the number of clusters and $B$ is the batch size. For large values of $K$, we achieve tractability by sampling $K$ random pairs of clusters every iteration instead of all $\binom{K}{2}$ possible pairs.

## 4 RELATED WORK

Majority of the work in survival analysis has dealt with the task of predicting the time to an observable terminal event (e.g., death), especially when the number of features is much larger than the number of subjects (Witten and Tibshirani, 2010a; pre; Hothorn et al., 2006; Shivaswamy et al., 2007). Recently, many deep learning approaches (Luck et al., 2017; Katzman et al., 2018; Lee et al., 2018; Ren et al., 2018) have been proposed for predicting the lifetime distribution of a subject given her covariates, while effectively handling censored data that typically arise in survival tasks. DeepHit (Lee et al., 2018) introduced a novel architecture and a ranking loss function in addition to the log-likelihood loss for lifetime prediction in the presence of multiple competing risks. Using a log-likelihood loss similar to DeepHit, Ren et al. (2018) propose a recurrent architecture to predict the survival distribution that captures sequential dependencies between neighbouring time points. Finally, Chapfuwa et al. (2018) introduce an adversarial learning framework to model the lifetime given the subject covariates. In contrast to these works on predicting lifetimes, our task is to cluster the subjects based on their underlying lifetime distributions.

There are relatively fewer works that perform lifetime clustering. Many unsupervised approaches have been proposed to identify cancer subtypes in gene expression data but do not consider the lifetime (Eisen et al., 1998; Alizadeh et al., 2000; Bhattacharjee et al., 2001; Sørlie et al., 2001; Bullinger et al., 2004), and may produce clusters that are entirely independent of the lifetimes. Semi-supervised clustering (Bair and Tibshirani, 2004) and supervised sparse clustering (Witten and Tibshirani, 2010b) use Cox scores (Cox, 1992) to identify features associated with the lifetime and treat these features differently while using k-means to perform the final clustering. Unlike these lifetime clustering methods, DeepCLife does not assume proportional hazards, and can smoothly handle the absence of termination signals. Our supplementary material has a more in-depth discussion of related work.

## 5 RESULTS

**Baselines.** We perform experiments on one synthetic dataset and two real-world datasets – **Friendster social network** and **MIMIC III healthcare dataset** (Johnson et al., 2016).

We compare the following lifetime clustering approaches: **(a) SSC-Bair**, a semi-supervised clustering method (Bair and Tibshirani, 2004) that performs k-means clustering on selected covariates that have high Cox scores (Cox, 1992); **(b) SSC-Gaynor** or supervised sparse clustering (Gaynor and Bair, 2013), a modification of sparse clustering (Witten and Tibshirani, 2010b) that weights the covariates based on their Cox scores; **(c) DeepHit+GMM**, a Gaussian mixture model applied over last layer embeddings learnt by DeepHit (Lee et al., 2018); **(d) DeepCLife-MMD**, the DeepCLife model with Maximum Mean Discrepancy (Gretton et al., 2012) as the divergence measure; **(e) DeepCLife-KuiperUB**, the DeepCLife model with the proposed divergence measure based on the Kuiper p-value upper bound (Equation (6)).

**Termination signals for evaluation and baselines.** (Timeout.) The competing methods and most evaluation metrics for survival applications require clear termination signals. In scenarios where we do not observe termination signals (e.g., Friendster experiments), we specify termination signals artificially when training the baselines using a pre-defined "timeout", i.e., $\beta^{(u)}(W_{fixed}) = \mathbf{1}[\chi^{(u)} > W_{fixed}]$. During evaluation, we use the same $W_{fixed}$ to specify the termination signals and compute the metrics. This helps the competing methods since they are trained and evaluated using termination signals with the same $W_{fixed}$, whereas our approach is not trained with these termination signals.

Table 2: **(Synthetic)** C-index (%) and Adjusted Rand index (%) for clusters with standard errors in parentheses for different methods [5].

| Method | $\mathcal{D}_{\{C_1,C_2\}}$ | | $\mathcal{D}_{\{C_1,C_3\}}$ | | $\mathcal{D}_{\{C_1,C_2,C_3\}}$ | |
|---|---|---|---|---|---|---|
| | C-index ↑ (%) | Adj. Rand Index ↑ (%) | C-index ↑ (%) | Adj. Rand Index ↑ (%) | C-index ↑ (%) | Adj. Rand Index ↑ (%) |
| SSC-Bair | 62.75 (0.35) | 74.66 (0.48) | 62.99 (0.26) | 56.86 (0.63) | 63.77 (0.24) | 47.67 (0.24) |
| SSC-Gaynor | 56.34 (0.50) | 19.88 (0.51) | 57.21 (0.43) | 16.60 (0.56) | 56.75 (0.32) | 5.84 (0.11) |
| DeepHit+GMM | 63.31 (0.32) | 85.05 (1.01) | 65.23 (0.21) | 78.47 (0.94) | 59.59 (1.98) | 38.77 (7.16) |
| DeepCLife-MMD | **64.35 (0.33)** | **98.47 (0.28)** | **67.25 (0.26)** | **99.68 (0.16)** | 62.17 (0.71) | 36.75 (1.10) |
| DeepCLife-KuiperUB | **64.37 (0.32)** | **99.02 (0.14)** | **67.24 (0.26)** | **99.94 (0.06)** | **68.96 (0.38)** | **73.61 (0.62)** |

Table 3: **(Friendster)** C-index (%), Integrated Brier Score (%) and Logrank score with standard errors in parentheses for different methods[4] and $K=2, 4$ clusters with number of training examples $N^{(\text{tr})}=10^5$.

| Method | $K=2$ | | | $K=4$ | | |
|---|---|---|---|---|---|---|
| | C-index ↑ (%) | I.B.S ↓ (%) | Logrank Score ↑ | C-index ↑ (%) | I.B.S. ↓ (%) | Logrank Score ↑ |
| SSC-Bair | 64.42 (0.15) | 22.18 (0.02) | 5479.27 ( 38.06) | 67.18 (0.13) | 21.55 (0.03) | 13013.86 ( 182.07) |
| SSC-Gaynor | 64.42 (0.18) | 22.17 (0.02) | 5557.41 ( 38.43) | 69.99 (0.28) | 21.62 (0.01) | 15204.81 ( 41.29) |
| DeepHit+GMM | 64.33 (1.80) | 22.04 (0.23) | 9207.46 (5031.48) | **76.55 (0.12)** | 20.64 (0.02) | 40703.01 ( 477.25) |
| DeepCLife-MMD | 67.49 (0.11) | 22.07 (0.02) | 27642.60 (1301.85) | 70.93 (1.80) | 22.40 (0.07) | 33030.93 (4277.24) |
| DeepCLife-KuiperUB | **75.58 (0.15)** | **20.13 (0.02)** | **47837.25 ( 297.63)** | **77.04 (0.88)** | **18.99 (0.20)** | **59236.36 (2126.55)** |

**Metrics.** Given the cluster assignments $\kappa(u') \in \{1, \ldots, K\}$ and the termination signals (possibly using a "timeout" window) for all the users $u'$ in the test data, we can obtain the empirical lifetime distribution of all the clusters $\hat{S}_k, \forall k \in \{1, \ldots, K\}$ using the Kaplan-Meier estimates (over the test data alone). Then, the empirical lifetime distribution of a user $u'$ is given by that of her assigned cluster, i.e., $\hat{S}^{(u')} := \hat{S}_{\kappa(u')}$. We use the following metrics[4] for evaluating the clusters obtained from the methods.

- **Logrank Score ↑.** Logrank test (Mantel, 1966) statistic is a non-parametric test that outputs high values when it is unlikely for the $K$ groups to have the same lifetime distribution.

- **Adjusted Rand index ↑.** The Adjusted Rand index (Hubert and Arabie, 1985) is a measurement of cluster agreement, compared to the ground truth clustering (if available). ARI is 0.0 for random cluster assignments and 1.0 for perfect assignments.

- **C-Index ↑.** Concordance index (Harrell et al., 1982) is a commonly used metric that calculates the fraction of pairs of subjects for which the model predicts the correct order of survival while also incorporating censoring. We use the expected lifetime obtained from the lifetime distribution $\hat{S}^{(u')}$ as the predicted lifetime of user $u'$. C-index is 1.0 for perfect predictions and 0.5 for random predictions.

- **Integrated Brier Score ↓.** Integrated Brier score (Brier, 1950; Graf et al., 1999) computes mean squared difference between the survival probabilities and the actual outcome over $[0, t_{\max}]$. It ranges from 0.25 for random predictions to 0 for perfect predictions.

Although cluster evaluation metrics like Logrank score are more suitable for the lifetime clustering task, we also use predictive measures like C-index and Brier Score to further validate the clusters.

**Evaluation.** We evaluate the models using 5-fold cross validation. We use the $i$th fold for testing and sample $N^{(\text{tr})}$ subjects from the remaining 4 folds for training. We use 20% of the $N^{(\text{tr})}$ subjects as validation for early stopping and hyperparameter tuning for the different approaches.

### 5.1 EXPERIMENTS

**Synthetic experiment.** We test our method with a synthetic dataset[6] for which we have the true clusters as ground truth. We generate 3 clusters $C_1, C_2$ and $C_3$ with different lifetime distributions such that $C_2$ and $C_3$ have proportional hazards, but lifetime distribution of $C_1$ crosses the other two curves (shown in Figure 2b). We choose an arbitrary time of measurement, $t_{\text{m}}=150$, to imitate right censoring. We sample $10^4$ subjects for each cluster, their features drawn from mixture of Gaussians. The lifetime $T^{(u)}$ of a subject $u$ is randomly sampled from the lifetime distribution of her ground truth cluster. Table 2 shows performance of the methods on three synthetic datasets: $\mathcal{D}_{\{C_1,C_2\}}, \mathcal{D}_{\{C_1,C_3\}}$,

---

[4]↑ indicates higher is better, ↓ indicates lower is better. Detailed description in the supplementary material.
[4]The values in bold are statistically significant with p-value $< 0.01$ using paired t-test over validation folds.
[5]More details about all the datasets and preprocessing steps can be found in the supplementary material.

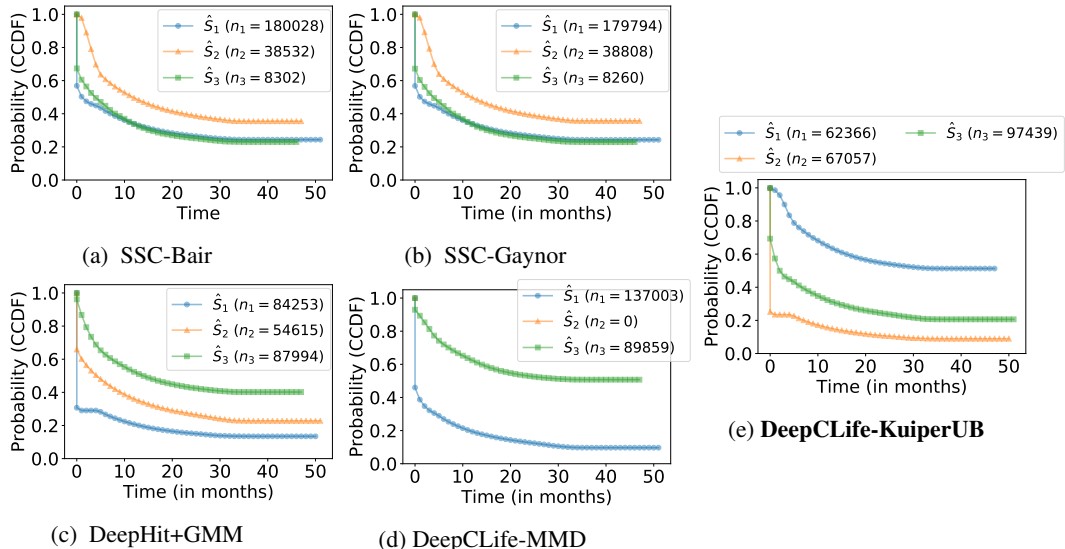

(a) SSC-Bair      (b) SSC-Gaynor

(c) DeepHit+GMM      (d) DeepCLife-MMD      (e) **DeepCLife-KuiperUB**

Figure 3: **(Friendster)** Empirical lifetime distributions of clusters obtained from different methods for $K{=}3$ (legend shows cluster sizes $n_1, n_2, n_3$). Baseline methods **(a-c)** employ a two-stage clustering process and do not guarantee clusters with maximally different lifetime distributions. **(d)** DeepCLife-MMD suffers from sample anomalies ($n_2 = 0$). **(e) DeepCLife-KuiperUB obtains clusters with significantly different lifetime distributions (best Logrank scores).**

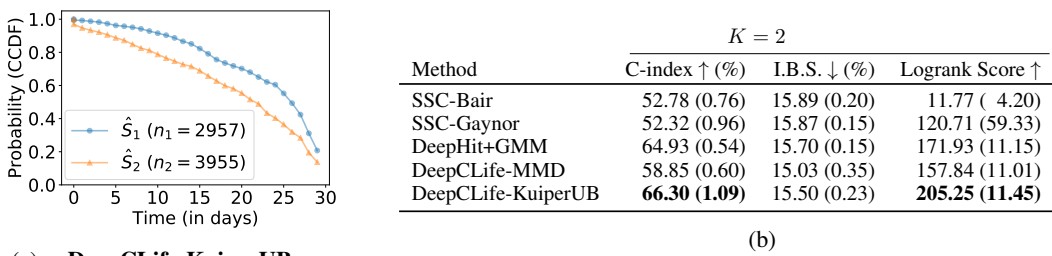

(a) **DeepCLife-KuiperUB**

|  | $K = 2$ | | |
| Method | C-index $\uparrow$ (%) | I.B.S. $\downarrow$ (%) | Logrank Score $\uparrow$ |
| --- | --- | --- | --- |
| SSC-Bair | 52.78 (0.76) | 15.89 (0.20) | 11.77 ( 4.20) |
| SSC-Gaynor | 52.32 (0.96) | 15.87 (0.15) | 120.71 (59.33) |
| DeepHit+GMM | 64.93 (0.54) | 15.70 (0.15) | 171.93 (11.15) |
| DeepCLife-MMD | 58.85 (0.60) | 15.03 (0.35) | 157.84 (11.01) |
| DeepCLife-KuiperUB | **66.30 (1.09)** | 15.50 (0.23) | **205.25 (11.45)** |

(b)

Figure 4: **(MIMIC III) (a)** Empirical lifetime distributions obtained by DeepCLife-KuiperUB for $K = 2$. **(b)** C-index (%), Brier Score (%) and Logrank score with standard errors for different methods on healthcare data for $K{=}2$.

$\mathcal{D}_{\{C_1,C_2,C_3\}}$. DeepCLife-KuiperUB and DeepCLife-MMD were able to recover perfect ground truth cluster assignments on $\mathcal{D}_{\{C_1,C_2\}}$ and $\mathcal{D}_{\{C_1,C_3\}}$, whereas the baseline methods performed invariably worse. On the harder dataset $\mathcal{D}_{\{C_1,C_2,C_3\}}$, DeepCLife-KuiperUB recovered the ground truth clusters almost twice as better than any other method.

**Friendster experiment.** Friendster dataset[5] consists of 15 million users in the Friendster online social network along with the comments sent and received by the users. We consider a subset with 1.1 million users who had participated in at least one comment. We use each user's profile information (like age, gender, location, etc.) as covariates, and define activity events as the comments sent or received by the user. We compute summary statistics of the activity events within $\tau = 5$ months from joining such as number of comments sent/received, number of people interacted with, mean inter-event times, etc. The task is to cluster new test users using their covariates and the summary statistics computed over initial $\tau = 5$ months of activity. Note that we do not observe *termination* signals (i.e., account deletion) for *any* subject in the data. We use an arbitrary window of $W_{\text{fixed}} = 10$ months over the inactivity period to obtain termination signals ($\approx 65\%$ assumed to have quit) for the competing methods and for computing the evaluation metrics; DeepCLife does not require such arbitrary specification but learns a smooth timeout window during training.

Table 3 shows results for $K{=}2, 4$ clusters. We note that the proposed method obtains higher C-index values and Brier scores compared to the baselines even without termination signals. DeepCLife-

KuiperUB achieves a significant improvement in Logrank scores compared to the baselines because its loss specifically maximizes for differences in empirical distributions, while also taking sample sizes into account. DeepCLife-MMD on the other hand does not account for sample sizes, and hence performs worse. The empirical lifetime distributions of the clusters obtained from different methods for $K=3$ are shown in Figure 3. The clusters obtained from the baselines **(a-c)** are not substantially different from each other. Although DeepCLife-MMD obtains clusters with distinct lifetime distributions, it suffers from sample anomalies i.e., outputs clusters with very few or no subjects (e.g., $\hat{S}_2$ in Figure 3d). DeepCLife-KuiperUB outputs clusters that have significantly different lifetime distributions with the best Logrank scores. Corresponding plots of empirical lifetime distributions for $K=2, 4, 5$ are presented in the supplementary material. For $K=4$, we observe that DeepCLife-KuiperUB finds crossing yet distinct lifetime distributions (see Figure 7e in the supplementary material). We present a heuristic to select optimal $K$ based on the validation performance of the clusters in Appendix A.4.

*Qualitative Analysis*: In the clusters found by DeepCLife-KuiperUB in Friendster, we see that a user in a *low-risk* cluster has on average 7.76 friends, sends 5.06 comments with an average response time of 20 days. On the other hand, a user in a *high-risk* cluster has just 1.56 friends on average and sends far fewer comments, around 1.07, but with a fast response time of 1.32 days. Interestingly, users that stayed longer in the system had lower activity rate in the beginning.

**MIMIC III experiment.** MIMIC III dataset[5] (Johnson et al., 2016) consists of around 46500 patients admitted to the Intensive Care Unit (ICU). The task is to cluster the patients based on their mortality within 30 days of admission to the ICU (Purushotham et al., 2017). Unlike Friendster experiments, we can observe terminal events. Lifetime of a patient is right-censored if she was discharged within the 30-day period ($\approx 84\%$ right-censored). We use only the initial $\tau=24$ hours of patient measurements (e.g. heart rate, respiratory rate, etc.) to perform the clustering. We compute summary statistics for each type of measurement, for example, number of heart rate measurements taken, mean heart rate, etc. to make the inputs compatible with the feedforward architecture in Figure 2a. Results for $K=2$ in Table 4b show that DeepCLife-KuiperUB achieves significantly better C-index and Logrank scores, followed by DeepHit+GMM. The corresponding empirical lifetime distributions of the clusters are shown in Figure 4a.

## 6 CONCLUSION

In this work we introduced Kuiper-based nonparametric loss function to maximize the divergence between empirical distributions, and a corresponding upper bound with easy-to-compute gradients. The loss function is then used to train a feedforward neural network to inductively map subjects into $K$ lifetime-based clusters without requiring *termination* signals. We show that this approach produces clusters with better C-index values and Logrank scores than competing methods.

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

# A    APPENDIX

## A.1    PROOF OF PROPOSITION 1

**Proposition** (Proposition 1). *Given the training data $\mathcal{D}$, a cluster $k$, the cluster assignment probabilities $\{\alpha_k^{(u)}(W_1)\}_{u\in\mathcal{D}}$, and the probabilities of termination $\{\beta^{(u)}(W_2)\}_{u\in\mathcal{D}}$, the maximum likelihood estimate of the empirical lifetime distribution of cluster $k$ is given by,*

$$\hat{S}_k(W_1, W_2; \mathcal{D})[t] = \prod_{j=0}^{t} \frac{s_k(W_1; \mathcal{D})[j] - d_k(W_1, W_2; \mathcal{D})[j]}{s_k(W_1; \mathcal{D})[j]} ,$$

*for all $t \in \{0, 1, \ldots, t_{max}\}$, where, $s_k(W_1; \mathcal{D})[j] = \sum_{u\in\mathcal{D}} \mathbf{1}[H^{(u)} \geq j] \cdot \alpha_k^{(u)}(W_1)$, is the expected number of subjects in cluster $k$ who are at risk (of termination) at time $j$, and, $d_k(W_1, W_2; \mathcal{D})[j] = \sum_{u\in\mathcal{D}} \mathbf{1}[H^{(u)} = j] \cdot \beta^{(u)}(W_2) \cdot \alpha_k^{(u)}(W_1)$, is the expected number of subjects that are predicted to have had a termination event at time $j$.*

*Proof.* For a subject $u$, let $e^{(u)} \sim$ Bernoulli$(\beta^{(u)}(W_2))$ be the sampled termination signal, i.e., $e^{(u)} = 1$ indicates that the subject's observed lifetime is her true lifetime, $H^{(u)} = T^{(u)}$. Let $\vec{\kappa}^{(u)} \sim$ Categorical$(\alpha_1^{(u)}(W_1), \ldots, \alpha_K^{(u)}(W_1))$ be the sampled cluster assignment of the subject.

Finally, let the empirical lifetime PMF of a cluster $k \in \{1, 2, \ldots K\}$ be defined as $P(T_k = t) := \lambda_{k,t}$ for $t \in \{0, 1, \ldots t_{\max}\}$, where $T_k$ is the true lifetime of a random subject in cluster $k$.

The likelihood of the training data $\mathcal{D}$ for a particular cluster $k$ given the sampled termination signal $\{e^{(u)}\}_{u\in\mathcal{D}}$ and the sampled cluster assignments $\{\vec{\kappa}^{(u)}\}_{u\in\mathcal{D}}$ can be written using Kaplan and Meier (1958) as,

$$L_k = \left[ \prod_{u\in\mathcal{D}} P(T_k = H^{(u)})^{e^{(u)}} P(T_k > H^{(u)})^{1-e^{(u)}} \right]^{\kappa_k^{(u)}} .$$

Next, we take expectation over the cluster assignments and termination signals to get,

$$L_k = \mathbb{E}_{\{\vec{\kappa}^{(u)}\}_u} \mathbb{E}_{\{e^{(u)}\}_u} \left[ \prod_{u\in\mathcal{D}} P(T_k = H^{(u)})^{e^{(u)}} P(T_k > H^{(u)})^{1-e^{(u)}} \right]^{\kappa_k^{(u)}} . \tag{7}$$

Using mean-field approximation and proceeding as Kaplan and Meier (1958) to write Equation (7) as a product over time instead of subjects, we get,

$$L_k = \left[ \prod_{j=0}^{t_{\max}} \lambda_{k,j}^{d_k[j]} (1 - \lambda_{k,j})^{s_k[j] - d_k[j]} \right] , \tag{8}$$

where, we have used $P(T_k = j) = \lambda_{k,j}$, $s_k[j]$ is a shorthand for $s_k(W_1, W_2; \mathcal{D})[j] = \sum_{u\in\mathcal{D}} \mathbf{1}[H^{(u)} \geq j] \cdot \alpha_k^{(u)}(W_1)$, the expected number of subjects at risk at time $j$, and, $d_k[j]$ is a shorthand for $d_k(W_1, W_2; \mathcal{D})[j] = \sum_{u\in\mathcal{D}} \mathbf{1}[H^{(u)} = j] \cdot \beta^{(u)}(W_2) \cdot \alpha_k^{(u)}(W_1)$, the expected number of subjects that are predicted to have had a termination event at time $j$.

Finally, maximizing the likelihood in Equation (8) with respect to $\lambda_{k,t}$ for all $k \in \{1, \ldots K\}$ and $t \in \{0, 1, \ldots t_{\max}\}$, and computing the lifetime distributions (CCDF) $\hat{S}_k$ from the corresponding lifetime PMFs finishes the proof. $\square$

## A.2    PROOF OF PROPOSITION 2

**Proposition** (Proposition 2). *Given two empirical lifetime distributions $\hat{S}_a$ and $\hat{S}_b$ with discrete support and sample sizes $n_a$ and $n_b$ respectively, define the maximum positive and negative separations between them as,*

$$\hat{D}_{a,b}^+ = \sup_{t \in \{0, \ldots t_{max}\}} (\hat{S}_a[t] - \hat{S}_b[t]) ,$$

$$\hat{D}_{a,b}^- = \sup_{t \in \{0, \ldots t_{max}\}} (\hat{S}_b[t] - \hat{S}_a[t]) .$$

*The Kuiper test $p$-value Kuiper (1960) gives the probability that $\Lambda$, the empirical deviation for $n_a$ and $n_b$ observations under the null hypothesis $S_a = S_b$ , exceeds the observed value $V = \hat{D}_{a,b}^+ + \hat{D}_{a,b}^-$:*

$$KD(\hat{S}_a, \hat{S}_b) \equiv P[\Lambda > V] = 2 \sum_{j=1}^{\infty} (4j^2 \lambda_{a,b}^2 - 1) e^{-2j^2 \lambda_{a,b}^2}, \tag{9}$$

*where $\lambda_{a,b} = \left( \sqrt{M_{a,b}} + 0.155 + \frac{0.24}{\sqrt{M_{a,b}}} \right) V$ and $M_{a,b} = \frac{n_a n_b}{n_a + n_b}$ is the effective sample size.*

*Then, the lower and upper bounds for the Kuiper p-value are,*

$$\frac{KD(\hat{S}_a, \hat{S}_b)}{2} \geq \mathbf{1}[r_{a,b}^{(lo)} \geq 1] \cdot \left( w(r_{a,b}^{(lo)} - 1, \lambda_{a,b}) \right.$$
$$\left. + v(r_{a,b}^{(lo)}, \lambda_{a,b}) \right) + v(r_{a,b}^{(up)}, \lambda_{a,b}) + w(r_{a,b}^{(up)} + 1, \lambda_{a,b}) ,$$

*and*

$$\frac{KD(\hat{S}_a, \hat{S}_b)}{2} \leq \min \left( \frac{1}{2}, \mathbf{1}[r_{a,b}^{(lo)} \geq 1] \cdot \left( w(r_{a,b}^{(lo)}, \lambda_{a,b}) \right. \right.$$
$$- w(1, \lambda_{a,b}) + v(r_{a,b}^{(lo)}, \lambda_{a,b}))$$
$$\left. + v(r_{a,b}^{(up)}, \lambda_{a,b}) - w(r_{a,b}^{(up)}, \lambda_{a,b}) \right), \tag{10}$$

*where $v(r, \lambda) = (4r^2\lambda^2 - 1)e^{-2r^2\lambda^2}$, with $w(r, \lambda) = -re^{-2r^2\lambda^2}$, and $r_{a,b}^{(lo)} = \lfloor \frac{1}{\sqrt{2}\lambda_{a,b}} \rfloor$, $r_{a,b}^{(up)} = \lceil \frac{1}{\sqrt{2}\lambda_{a,b}} \rceil$.*

*Proof.* To prove these bounds, we first regard that the indefinite integral $w(r, \lambda) = \int v(r, \lambda) dr = -re^{-2r^2\lambda^2} + C$. Then, for a given $\lambda$, $v(r, \lambda)$ is monotonically increasing w.r.t $r$ in the interval $(0, \frac{1}{\sqrt{2}\lambda})$ and monotonically decreasing w.r.t $r$ in the interval $(\frac{1}{\sqrt{2}\lambda}, \infty)$.

Let $r^{(lo)} = \lfloor \frac{1}{\sqrt{2}\lambda} \rfloor$, $r^{(up)} = \lceil \frac{1}{\sqrt{2}\lambda} \rceil$ and $J_{[l,m]}(\lambda) = \sum_{r=l}^{m} v(r, \lambda)$. Here, we assume that $r^{(lo)} \neq r^{(up)}$, i.e., $\frac{1}{\sqrt{2}\lambda}$ is not an integer. The bounds can be easily modified for when $r^{(lo)} = r^{(up)}$. Now, $v(r, \lambda)$ is decreasing w.r.t $r$ in $(r^{(up)}, \infty)$, and we have,

$$\int_{r^{(up)}+1}^{\infty} v(r, \lambda) dr \leq J_{[r^{(up)}+1, \infty]}(\lambda) \leq \int_{r^{(up)}}^{\infty} v(r, \lambda) dr ,$$

and adding $v(r^{(up)}, \lambda)$ throughout we get,

$$v(r^{(up)}, \lambda) + \int_{r^{(up)}+1}^{\infty} v(r, \lambda) dr$$
$$\leq J_{[r^{(up)}, \infty]}(\lambda) \leq v(r^{(up)}, \lambda) + \int_{r^{(up)}}^{\infty} v(r, \lambda) dr . \tag{11}$$

Similarly, since $v(r, \lambda)$ is increasing in $(0, r^{(lo)})$, we have,

$$\int_0^{r^{(lo)}-1} v(r, \lambda) dr \leq J_{[1, r^{(lo)}-1]}(\lambda) \leq \int_1^{r^{(lo)}} v(r, \lambda) dr ,$$

and adding $v(r^{(lo)}, \lambda)$ throughout we get,

$$\int_0^{r^{(lo)}-1} v(r, \lambda) dr + v(r^{(lo)}, \lambda)$$
$$\leq J_{[1, r^{(lo)}]}(\lambda) \leq \int_1^{r^{(lo)}} v(r, \lambda) dr + v(r^{(lo)}, \lambda) , \tag{12}$$

where we assume that $r^{(lo)} \geq 1$, otherwise, $J_{[1, r^{(lo)}]}(\lambda)$ is trivially zero.

Combining equations equation 11 and equation 12, we get,

$$\mathbf{1}[r^{(lo)} \geq 1] \cdot \left( \int_0^{r^{(lo)}-1} v(r,\lambda)dr + v(r^{(lo)},\lambda) \right)$$

$$+ v(r^{(up)},\lambda) + \int_{r^{(up)}+1}^{\infty} v(r,\lambda)dr$$

$$\leq \quad \mathbf{1}[r^{(lo)} \geq 1] \cdot J_{[1,r^{(lo)}]}(\lambda) + J_{[r^{(up)},\infty]}(\lambda) \quad \leq$$

$$\mathbf{1}[r^{(lo)} \geq 1] \cdot \left( \int_1^{r^{(lo)}} v(r,\lambda)dr + v(r^{(lo)},\lambda) \right)$$

$$+ v(r^{(up)},\lambda) + \int_{r^{(up)}}^{\infty} v(r,\lambda)dr \ . \tag{13}$$

We use the fact that $\text{KD}(\hat{S}_a, \hat{S}_b) = 2 \cdot J_{[1,\infty]}(\lambda_{a,b})$, $J_{[1,\infty]}(\lambda) = \mathbf{1}[r^{(lo)} \geq 1] \cdot J_{[1,r^{(lo)}]}(\lambda) + J_{[r^{(up)},\infty]}(\lambda)$, and $w(r,\lambda) = \int v(r,\lambda)dr$, and that $\text{KD}(\hat{S}_a, \hat{S}_b) \leq 1$ to complete the proof. □

## A.3 ADDITIONAL RESULTS

### A.3.1 FRIENDSTER EXPERIMENT

Results on the Friendster experiments for $K = 3, 5$ are shown in Table 4. We observe that DeepCLife outperforms all baselines by large margins, particularly in the Logrank score. Figures 5 to 8 show empirical lifetime distributions of the clusters found by the methods on the Friendster dataset for $K = 2, 3, 4, 5$ respectively. Baseline methods SSC-Bair, SSC-Gaynor and DeepHit+GMM employ a two-stage clustering process and do not guarantee clusters with maximally different lifetime distributions. DeepCLife-MMD does not account for sample sizes of empirical distributions and suffers from sample anomalies, i.e., outputs clusters containing almost no subjects. DeepCLife-KuiperUB obtains clusters with the best Logrank scores. Interestingly, DeepCLife finds clusters with crossing yet distinct lifetime distributions for $K = 4$ (see Figure 7e).

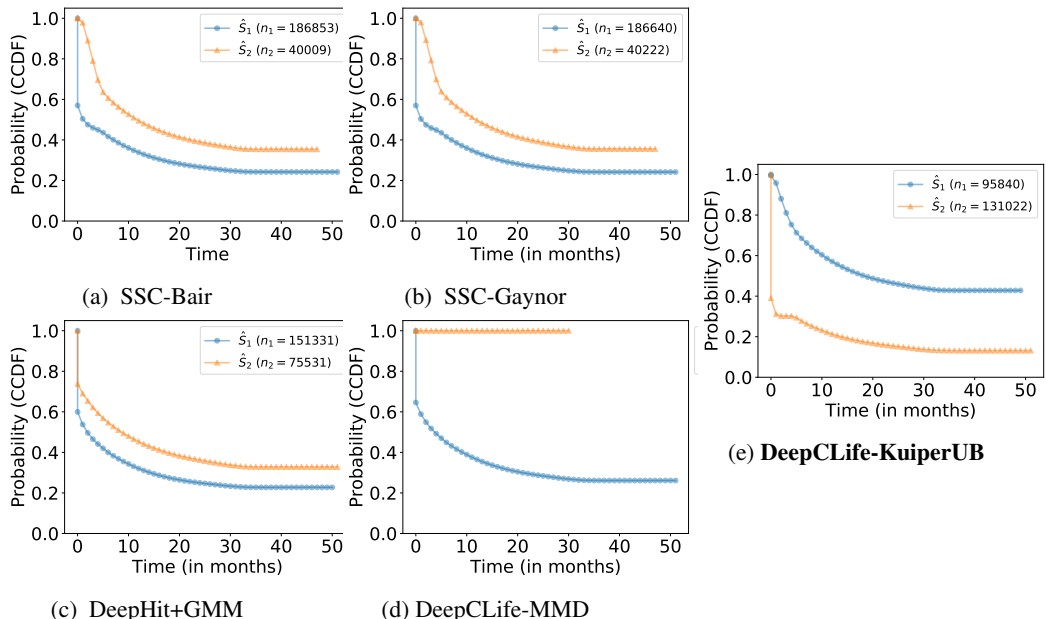

Figure 5: **(Friendster)** Empirical lifetime distributions of clusters obtained from different methods for $K=2$ (legend shows cluster sizes $n_1, n_2$).

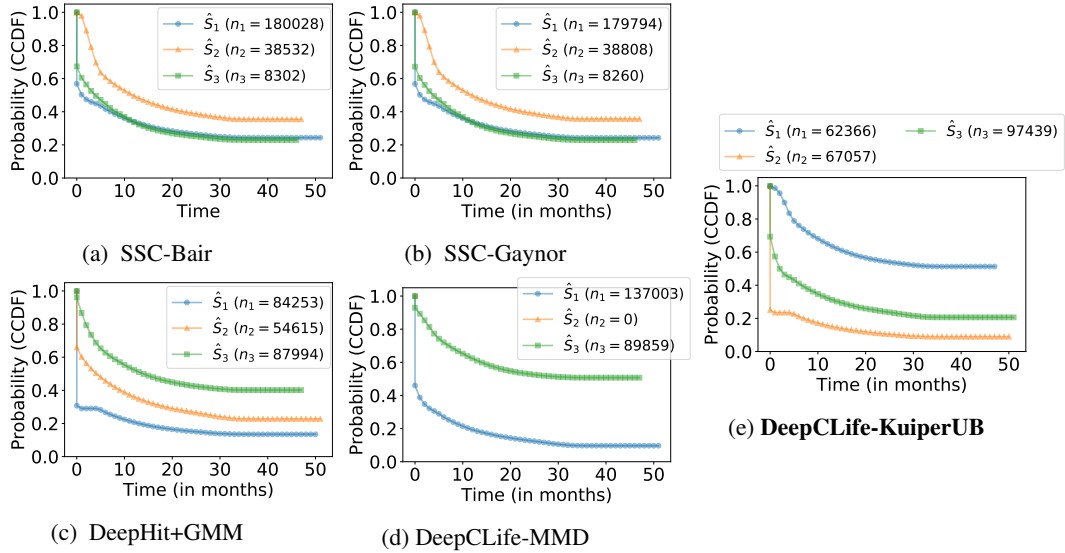

Figure 6: **(Friendster)** Empirical lifetime distributions of clusters obtained from different methods for $K=3$ (legend shows cluster sizes $n_1, n_2, n_3$).

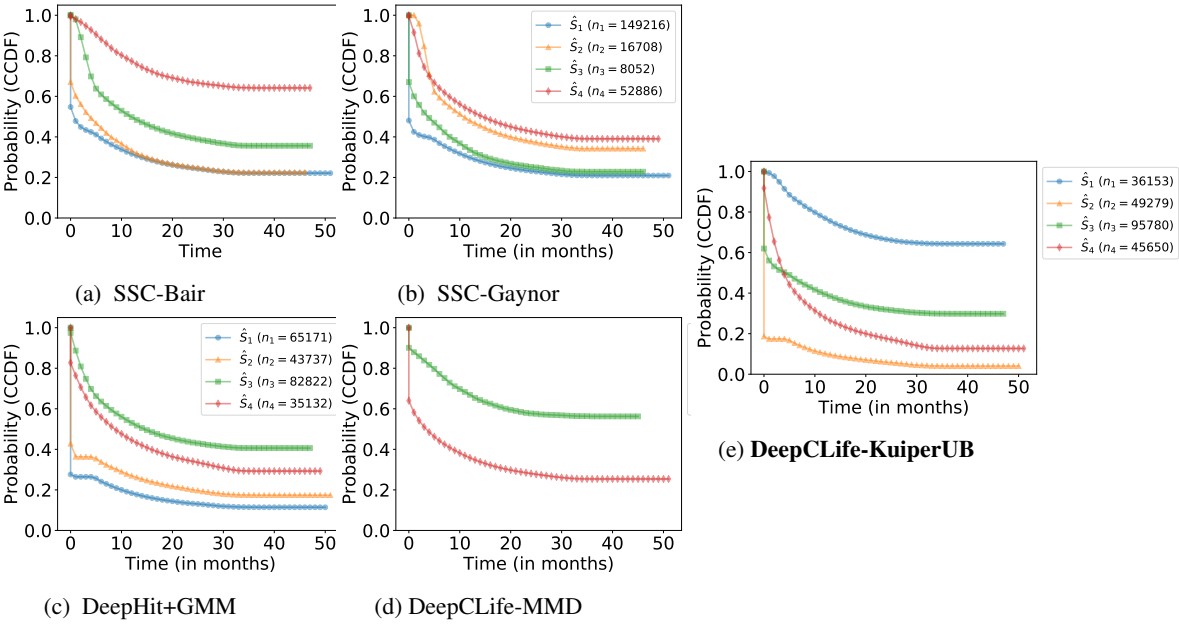

Figure 7: **(Friendster)** Empirical lifetime distributions of clusters obtained from different methods for $K=4$ (legend shows cluster sizes).

### A.3.2 MIMIC III EXPERIMENT

Figure 9 shows the empirical lifetime distributions of the clusters found by DeepCLife-KuiperUB in MIMIC III dataset for $K = 2 \dots 6$.

### A.4 OPTIMAL NUMBER OF CLUSTERS : $K$

Similar to standard clustering algorithms, choice of $K$ is subjective and depends on the downstream application of the clusters. Here we discuss a heuristic to use Logrank scores as a useful guide to choose the optimal number of clusters. For the Friendster experiments, Figure 10 shows the Logrank

Table 4: **(Friendster)** C-index (%), Integrated Brier Score (%) and Logrank score with standard errors in parentheses for different methods[3] and $K=3, 5$ clusters with number of training examples $N^{(\text{tr})}=10^5$.

| | K = 3 | | | K = 5 | | |
|---|---|---|---|---|---|---|
| Method | C-index ↑ (%) | I.B.S ↓ (%) | Logrank Score ↑ | C-index ↑ (%) | I.B.S. ↓ (%) | Logrank Score ↑ |
| SSC-Bair | 64.23 (0.24) | 22.19 (0.02) | 5277.82 ( 43.63) | 75.80 (0.16) | 20.26 (0.02) | 43605.16 ( 91.39) |
| SSC-Gaynor | 64.13 (0.21) | 22.18 (0.02) | 5400.75 ( 39.59) | 75.66 (0.20) | 20.48 (0.01) | 41824.28 ( 151.85) |
| DeepHit+GMM | 74.81 (0.11) | 20.97 (0.06) | 33880.55 ( 426.74) | 76.17 (0.27) | 20.54 (0.11) | 39254.67 (1721.56) |
| DeepCLife-MMD | 72.75 (1.24) | **18.94 (0.65)** | 50173.94 (4809.80) | 63.08 (2.69) | 21.55 (0.54) | 14308.90 (9149.41) |
| DeepCLife-KuiperUB | **78.48 (0.32)** | 19.70 (0.55) | 54191.28 ( 436.72) | 76.51 (1.25) | **18.99 (0.16)** | 56130.21 (4036.02) |

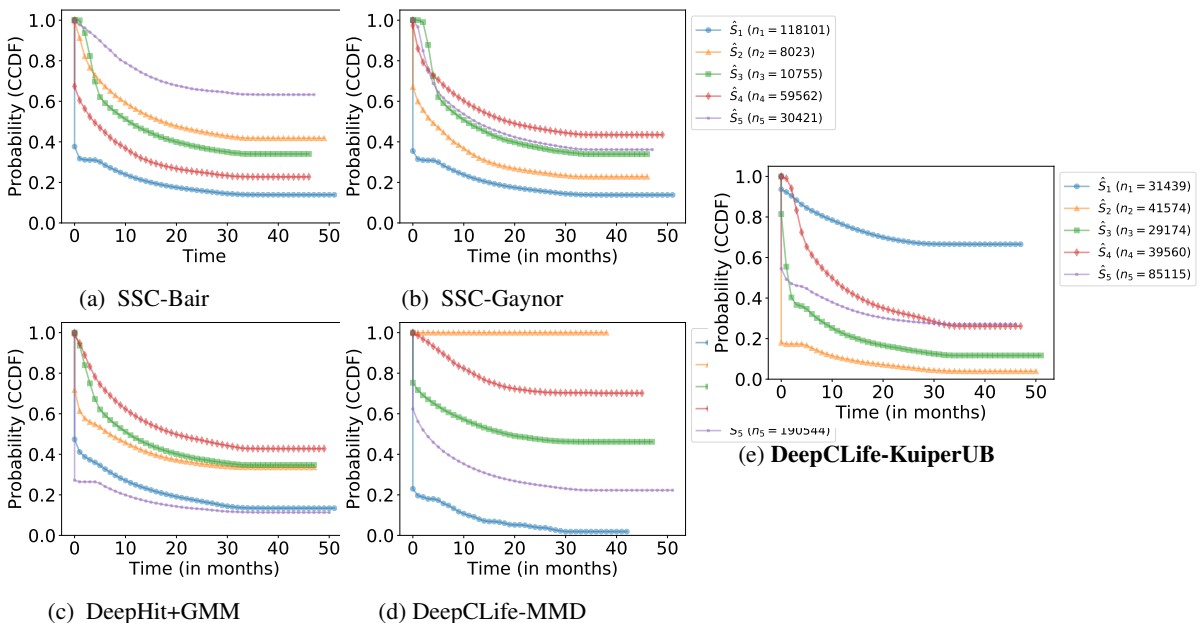

Figure 8: **(Friendster)** Empirical lifetime distributions of clusters obtained from different methods for $K=5$ (legend shows cluster sizes).

scores on a held-out validation set for clusters obtained from DeepCLife-KuiperUB with $K = 2\ldots 7$. We observe that $K = 4$ gives the best mean Logrank score and $K = 5$ is close behind. However, for $K > 5$, the Logrank scores drop drastically indicating that there are no more clusters that have distinct lifetime distributions from the ones already found.

Similarly for the MIMIC III experiments, Figure 11 shows the Logrank scores on a held-out validation set for clusters obtained from DeepCLife-KuiperUB with $K = 2\ldots 8$. We see that choosing $K = 6$ gives the highest Logrank score. However, we also observe that from $K = 2$ to $K = 5$ the scores do not increase significantly; a practitioner might decide to use $K = 2$ for this reason.

## A.5 CHOICE OF MINIMUM VS SUM IN EQUATIONS (1) AND (4)

We define the clustering problem in Definition 3 as finding a mapping $\kappa$ such that,

$$\kappa^\star = \arg\max_{\kappa \in \mathcal{K}} \min_{\substack{i,j \in \{1\ldots K\}, \\ i \neq j}} \Delta(\hat{S}_i(\kappa), \hat{S}_j(\kappa)),$$

where $\mathcal{K}$ is a set of all mappings, $\hat{S}_k(\kappa)$ is the empirical lifetime distribution of subjects in the training data $\mathcal{D}$ mapped to cluster $k$ through $\kappa$, and $\Delta$ is an empirical distribution divergence measure.

In this section, we discuss the issues that arise when one chooses to optimize the sum of divergences over all pairs of clusters rather than the minimum.

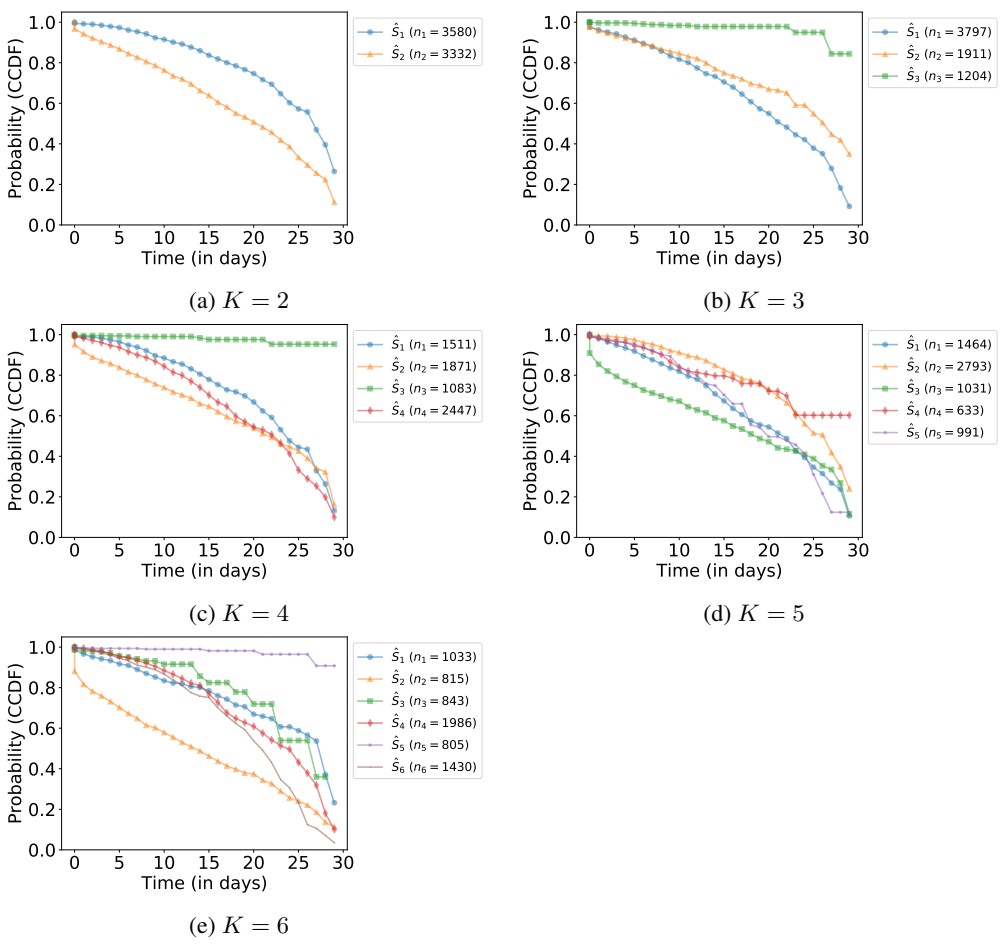

Figure 9: **(MIMIC III)** Empirical lifetime distributions of clusters obtained from **DeepCLife-KuiperUB** with $K=2\ldots6$ (legend shows cluster sizes).

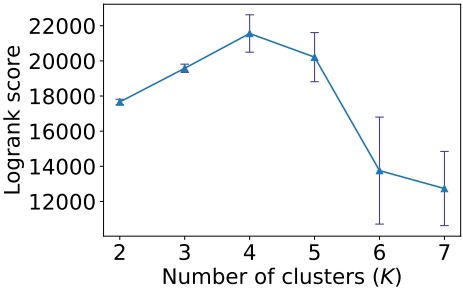

Figure 10: **(Friendster)** Logrank scores (error bars denote standard errors) on a held-out validation set for the clusters obtained from **DeepCLife-KuiperUB** with different values of $K$. $K = 4$ seems to give the best mean performance, while $K = 5$ is close behind; increasing $K$ further drastically reduces the Logrank score.

Let $K = 4$ and consider balanced clusters with lifetime distributions $\hat{S}_1 = \hat{S}_2 \neq \hat{S}_3 = \hat{S}_4$. Note that such a clustering is not desirable since there are virtually only 2 clusters found (as $\hat{S}_1$ and $\hat{S}_2$ are the same, etc). But for such an assignment, the sum of divergences is relatively high $\sum_{i \neq j} \Delta(\hat{S}_i, \hat{S}_j) = 4 \cdot \Delta(\hat{S}_1, \hat{S}_3)$. The sum could be further maximized by simply increasing

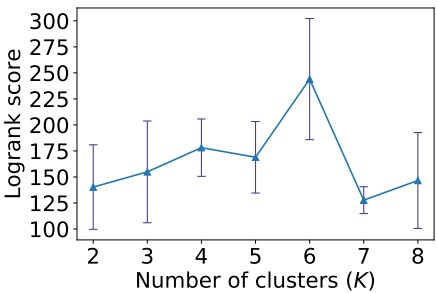

Figure 11: **(MIMIC III)** Logrank scores (error bars denote standard errors) on a held-out validation set for the clusters obtained from **DeepCLife-KuiperUB** with different values of $K$. $K = 6$ gives the best Logrank performance. However, Logrank scores for $K = 2$ to $K = 5$ do not increase significantly; suggesting $K = 2$ could be a good choice as well.

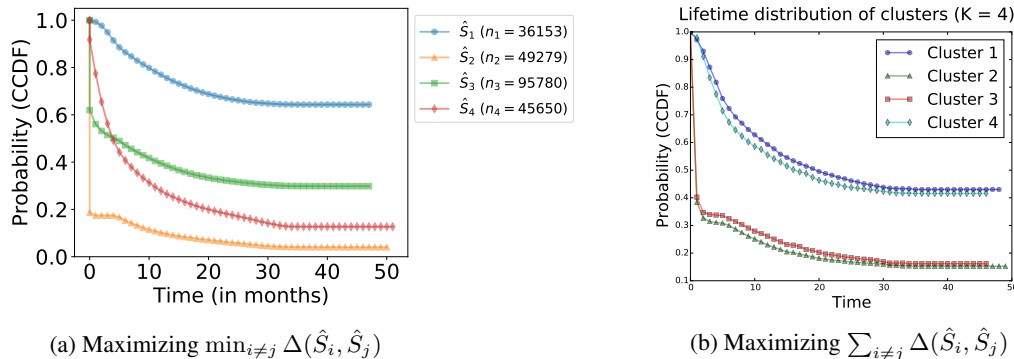

(a) Maximizing $\min_{i \neq j} \Delta(\hat{S}_i, \hat{S}_j)$      (b) Maximizing $\sum_{i \neq j} \Delta(\hat{S}_i, \hat{S}_j)$

Figure 12: **(Friendster)** Clusters found by the proposed approach with $K = 4$ when maximizing the minimum divergence (left) vs maximizing the sum of divergences (right).

divergence between $\hat{S}_1$ and $\hat{S}_3$ while keeping $\hat{S}_2 = \hat{S}_1$ and $\hat{S}_3 = \hat{S}_4$. On the other hand, since $\min_{i \neq j} \Delta(\hat{S}_i, \hat{S}_j) = 0$, the model should maximize the divergence between $\hat{S}_1$ and $\hat{S}_2$ first.

Figure 12 shows the clusters found by the proposed approach while maximizing the minimum divergence vs maximizing the sum of divergences in the Friendster dataset with $K = 4$. We observe that the clusters obtained using the sum of divergences as the loss exhibit the property discussed above (i.e., clusters 2 and 3 in Figure 12 are almost the same).

## A.6    ALGORITHM

For ease of use, we present Algorithm 1 to compute the proposed divergence measure $\log(\text{KD}^{(\text{up})}(\hat{S}_a, \hat{S}_b))$.

## A.7    TIME/SPACE COMPLEXITY OF DEEPCLIFE-KUIPERUB

Here we discuss the time and space complexity of the proposed algorithm. We only discuss the complexity of computation of the proposed loss function for one iteration assuming a minibatch of size $B$, number of clusters $K$ and discrete times with $0, 1, \ldots t_{\max}$. First, we can compute the empirical lifetime distributions of all the $K$ clusters in $O(KB)$ time using $O(Kt_{\max})$ space. Computing upper bound of the Kuiper p-value for two lifetime distributions using Algorithm 1 requires $O(t_{\max})$ time and $O(t_{\max})$ space. Since we compute the upper bound for all $\binom{K}{2}$ pairs, the time complexity then becomes $O(K^2 t_{\max})$ whereas the space complexity remains $O(t_{\max})$. The total time and space complexity of the proposed algorithm are $O(KB + K^2 t_{\max})$ and $O(Kt_{\max})$ respectively.

---

**Algorithm 1** Kuiper Divergence Upper Bound (Log)

---

**Require:** Two empirical lifetime distributions $\hat{S}_a$ and $\hat{S}_b$ with sample sizes $n_a$ and $n_b$ respectively

---

1: **function** KUIPERUB($\hat{S}_a, \hat{S}_b, n_a, n_b$)
2:      $D^+ \leftarrow \max(\hat{S}_a - \hat{S}_b)$                           ▷ $\hat{S}_a, \hat{S}_b$ are vectors of length $t_{\max}+1$
3:      $D^- \leftarrow \max(\hat{S}_b - \hat{S}_a)$
4:      $V \leftarrow D^+ + D^-$                                     ▷ Kuiper statistic
5:      $M \leftarrow \frac{n_a n_b}{n_a + n_b}$                                   ▷ Effective sample size
6:      $\lambda \leftarrow (\sqrt{M} + 0.155 + \frac{0.24}{\sqrt{M}})V$
7:      $r^{(\mathrm{lo})} \leftarrow \lfloor \frac{1}{\sqrt{2}\lambda} \rfloor$
8:      $r^{(\mathrm{up})} \leftarrow \lceil \frac{1}{\sqrt{2}\lambda} \rceil$
9:      **if** $r^{(\mathrm{lo})} \geq 1$ **then**
10:         KD-UB $\leftarrow \mathrm{W}(r^{(\mathrm{lo})}, \lambda) - \mathrm{W}(1, \lambda) + \mathrm{V}(r^{(\mathrm{lo})}, \lambda) + \mathrm{V}(r^{(\mathrm{up})}, \lambda) - \mathrm{W}(r^{(\mathrm{up})}, \lambda)$
11:      **else**
12:         KD-UB $\leftarrow \mathrm{V}(r^{(\mathrm{up})}, \lambda) - \mathrm{W}(r^{(\mathrm{up})}, \lambda)$
13:      **return** $\log$ KD-UB

14: **function** V($r, \lambda$)
15:      **return** $(4r^2\lambda^2 - 1)e^{-2r^2\lambda^2}$

16: **function** W($r, \lambda$)
17:      **return** $-re^{-2r^2\lambda^2}$

---

### A.7.1 TRACTABILITY THROUGH SAMPLING

Each iteration of optimization involves computing the Kuiper upper bound for all $\binom{K}{2}$ pairs of clusters, hence can be prohibitively expensive for large values of $K$. Rather, we propose to sample randomly $p = O(K)$ pairs without replacement from the $\binom{K}{2}$ possible pairs of clusters. The time and space complexity reduces to $O(KB + Kt_{\max})$ and $O(Kt_{\max})$ respectively.

### A.8 ADDITIONAL DETAILS ABOUT DATASETS

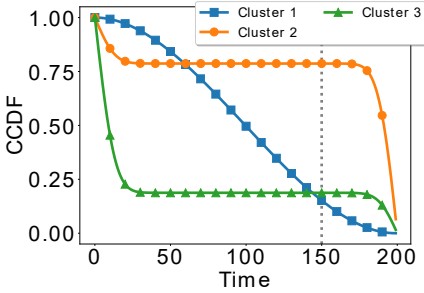

Figure 13: **(Toy)** True lifetime distributions of simulated clusters (with $t_{\mathrm{m}} = 150$).

**Synthetic dataset.** We also test our method with a synthetic dataset for which we have the true clustering as ground truth. We generate 3 clusters $C_1, C_2$ and $C_3$ with different lifetime distributions, as depicted in Figure 13. Note that while the lifetime distributions $C_2$ and $C_3$ follow proportional hazards assumption, the lifetime distribution of $C_1$ violates this assumption by crossing the other survival curves.

We simulate $10^4$ subjects for each cluster $k$, and generate 20 random features for each sampled subject. The lifetime $T^{(u)}$ of a subject $u$ in a cluster was randomly sampled from the corresponding

lifetime distribution of the cluster. We also choose an arbitrary time of measurement, $t_m = 150$, to imitate right censoring.

We generate covariates $X^{(u)}$ of each subject $u$ as two sets of 10 attributes. Each attribute is simulated from a mixture of three Gaussians with means $\mu_{k,i}^{(1)}$, $\mu_{k,i}^{(2)}$, and $\mu_{k,i}^{(3)}$, for $i = 1 \ldots 20$ and $k = 1, 2, 3$. For the first set of 10 features, these means are uniformly selected from the range $[0, 30]$ and a variance of 1 is used. For the second set of features, the three means are uniformly selected from the range $[0, 30]$ and a variance of 10 is used. Now that the modes of each Gaussian of each feature are defined, we generate the features as follows: for each feature $i$ of user $u$ of cluster $k$, we uniformly choose one of the three modes $m \in \{1, 2, 3\}$. Once the mode is selected, we generate a value from $\mathcal{N}(\mu_{k,i}^{(m)}, \sigma^2)$ (where $\sigma^2$ is 1 for $i \in [1, 10]$ and 10 for $i \in [11, 20]$). We decided to have these two sets of features as the first one (with lower variance) represents more relevant features which can help identify the clusters, while the second set (with higher variance) represents less meaningful features that may be similar for all clusters.

We create three separate datasets $\mathcal{D}_{\{C_1, C_2\}}$, $\mathcal{D}_{\{C_1, C_3\}}$ and $\mathcal{D}_{\{C_1, C_2, C_3\}}$ such that $\mathcal{D}_\mathcal{C}$ is a union of all the clusters in the set $\mathcal{C}$. We report the C-index and Adjusted Rand index for the proposed method and the baselines on these 3 simulated datasets in Table 2.

**Friendster dataset.** Friendster dataset consists of around 15 million users with 335 million friendship links in the Friendster online social network. Each user has profile information such as age, gender, marital status, occupation, and interests. Additionally, there are user comments on each other's profile pages with timestamps that indicate activity in the site.

In our experiments, we only use data from March 2002 to March 2008, as after March 2008 Friendster's monthly active users have been significantly affected with the introduction of "new Facebook wall" (Ribeiro and Faloutsos, 2015). From this, we only consider a subset of 1.1 million users who had participated in at least one comment, and had specified their basic profile information like age and gender. We will make our processed data available to the public at *location* (anonymized). We use each user's profile information (like age, gender, relationship status, occupation and location) as features. We convert the nominal attributes to one-hot encoded features. We compute the summary statistics of the user's activity over the initial $\tau = 5$ months such as the number of comments sent and received, number of individuals interacted with, etc. In total, we construct 60 numeric features that are used for each of the models in our experiments.

**MIMIC III dataset.** MIMIC III dataset consists of patients compiled from two ICU databases: CareVue and Metavision. The dataset consist of 50,000 patients with a total of 330 million different measurements recorded. We mainly use three tables in the dataset: `PATIENTS`, `ADMISSIONS`, `CHART_EVENTS`. `PATIENTS` table consists of demographic information about every patient such as gender, date of birth, and ethnicity. `ADMISSIONS` table consists of the admission and discharge times of the patients. A patient can have multiple admissions; in our experiments, we only consider their longest stay. Finally, `CHART_EVENTS` table records all the measurements of the patients taken during their stay such as daily weight, heart rate and respiratory rate. The measurements of the patients are shifted by an unknown offset (consistent across measurements for a single patient) for anonymity, hence all times are relative to the patient.

Each entry in the `CHART_EVENTS` table has a column for item identifier that specifies the measured item (like heart rate). Since the dataset is compiled from different databases, same item can have multiple identifiers. We use the following time series for features: 'Peak Insp. Pressure', 'Plateau Pressure', 'Respiratory Rate', 'Heart Rate', 'Mean Airway Pressure', 'Arterial Base Excess', 'BUN', 'Creatinine', 'Magnesium', 'WBC' and 'Hemoglobin'. We observe each of these time series for $\tau = 24$ hours from the patient's admission to the ICU, and compute summary statistics such as number of observations, mean, variance, mean inter-arrival times, etc. Complete processing script is provided in the supplementary material.

## A.9 Metrics

Here we describe the metrics used for evaluating the clusters obtained from the methods.

**Concordance Index.** Concordance index or C-index (Harrell et al., 1982) is a commonly used metric in survival applications (Alaa and van der Schaar, 2017; Luck et al., 2017) to quantify a model's ability to discriminate between subjects with different lifetimes. It calculates the fraction of pairs of subjects for which the model predicts the correct order of survival while also incorporating censoring. Concordance index can be seen as a generalization of AUROC (Area Under ROC curve) and can be interpreted in a similar fashion, i.e., a C-index score of 1 denotes perfect predictions as compared to a C-index score of 0.5 for random predictions.

**Integrated Brier Score.** Brier score (Brier, 1950; Graf et al., 1999) is a quadratic scoring rule, computed as follows for survival applications,

$$B(u) = \frac{1}{t_{\max} + 1} \sum_{t=0}^{t_{\max}} (\mathbf{1}[T^{(u)} > t] - \hat{S}^{(u)}[t])^2 \, , \tag{14}$$

where for a given subject $u$, $T^{(u)}$ is her true lifetime and $\hat{S}^{(u)}$ is her empirical lifetime CCDF. Brier score is 0 for perfect predictions and 0.25 for random predictions.

**Logrank Test Score.** Logrank test Mantel (1966) is a non-parametric hypothesis test that is used to compare survival distributions. The null hypothesis for the test is defined as $H_0 : f_1(t) = \ldots = f_K(t)$, i.e., the $K$ groups have the same survival distribution. It is most appropriate when the observations are censored and the censoring is independent of the events. High values of the logrank statistic denote that it is unlikely that the $K$ groups have the same survival distribution.

**Adjusted Rand index.** The Rand index (Rand, 1971) is a measurement of cluster agreement, compared to the ground truth clustering (if available). The adjusted version (Hubert and Arabie, 1985) is a corrected-for-chance version of the Rand index, which assigns a score of 0 for the expected Rand index, obtained by random cluster assignment. The maximum value of the Adjusted Rand index is 1. We use this metric only for the toy dataset where we have the ground truth cluster assignments available.

### A.10 Neural Network Hyperparameters

| Parameter | Values |
| --- | --- |
| nHiddenLayers | [1, 2, 3] |
| nHiddenUnits | [128, 256] |
| Minibatch Size | [128, 256, 1024] |
| Learning Rate | $[10^{-3}, 10^{-2}]$ |
| Activation | [Tanh, ReLU] |
| Batch normalization | [True, False] |
| L2 Regularization | $[10^{-2}, 0]$ |

Table 5: Different neural network hyperparameters for the proposed approach used in our experiments. The best set of hyperparameters was chosen based on validation performance.

## B Related Work

**Traditional survival analysis models.** The Cox regression model (Cox, 1992) is a widely used method in survival analysis to estimate the hazard function $\lambda^{(u)}(t) = \frac{dF^{(u)}(t)}{S^{(u)}(t)}$, where $dF^{(u)}$ is the probability density of $F^{(u)}$. The hazard function is then estimated using the covariates, $X^{(u)}$, of a subject $u$. The hazard function has the form, $\lambda(t|X^{(u)}) = \lambda_0(t) \cdot e^{\{\beta^T X^{(u)}\}}$, where $\lambda_0(t)$ is a base hazard function common for all subjects, and $\beta$ are the regression coefficients. The model assumes that the ratio of hazard functions of any two subjects is constant over time. This assumption is violated frequently in real-world datasets (Li et al., 2015a). A near-extreme case when this assumption does not hold is shown in Figure 2b, where the survival curves of two groups of subjects cross each other.

Majority of the work in survival analysis has dealt with the task of predicting the survival outcome especially when the number of features is much higher than the number of subjects (Witten and

Tibshirani, 2010a; pre; Hothorn et al., 2006; Shivaswamy et al., 2007). A number of approaches have also been proposed to perform feature selection in survival data (Ishwaran et al., 2010; Lagani and Tsamardinos, 2010). In the social network scenario, Sun et al. (2012) predicts the relationship building time, that is, the time until a particular link is formed in the network. Alaa and van der Schaar (2017) proposed a nonparametric Bayesian approach for survival analysis in the case of more than one competing events (multiple diseases). They not only assume the presence of termination signals but also the type of event that caused the termination.

**Survival (or lifetime) clustering.** There have been relatively fewer works that perform lifetime clustering. Many unsupervised approaches have been proposed to identify cancer subtypes in gene expression data without considering the survival outcome (Eisen et al., 1998; Alizadeh et al., 2000). Traditional semi-supervised clustering methods (Aggarwal et al., 2004; Basu et al., 2002; 2004; Nigam et al., 1998) do not perform well in this scenario since they do not provide a way to handle the issues with right censoring. Bair and Tibshirani (2004) proposed a semi-supervised method for clustering survival data in which they assign Cox scores (Cox, 1992) for each feature in their dataset and considered only the features with scores above a predetermined threshold. Then, an unsupervised clustering algorithm, like k-means, is used to group the individuals using only the selected features. Such an approach can miss out on clusters when the features are weakly associated with the survival outcome since such features are discarded immediately after the initial screening. In order to overcome this issue, Gaynor and Bair (2013) proposes *supervised sparse clustering* as a modification to the sparse clustering algorithm of Witten and Tibshirani (2010b). The sparse clustering algorithm has a modified $k$-means score that uses distinct weights in the feature set; it initializes these feature weights using Cox scores (Cox, 1992) and optimizes the same objective.

Bair and Tibshirani (2004) and Gaynor and Bair (2013) assume the presence of termination signals. Additionally, there is a disconnect between the use of survival outcomes and the clustering step in these algorithms. In this paper, we provide a loss function that quantifies the divergence between lifetime distributions of the clusters, and we minimize said loss function using a neural network in order to obtain the optimal clusters.

**Divergence measures.** Since we need to optimize the divergence between two distributions using a finite sample of training data, this problem is related to two-sample tests in statistics. Recently, there is growing interest in loss functions for comparing two distributions, including the Wasserstein distance (Dudley, 2002, p. 420) and the Maximum Mean Discrepancy (MMD) (Gretton et al., 2009; Fukumizu et al., 2008), specifically in the context of training generative adversarial networks (GANs) (Arjovsky et al., 2017; Li et al., 2015b).

Traditional two-sample tests, such as the Kolmogorov-Smirnov (KS) test, are generally disregarded in learning tasks due to their alternating infinite series that makes gradient computations computationally expensive. The alternative state-of-the-art divergence approach that seems most suited to our problem is MMD Gretton et al. (2009), unfortunately, as seeing in our results, they can be less resilient to small sample sizes in empirical distributions.

**Deep survival methods.** Many deep learning approaches Luck et al. (2017); Katzman et al. (2018); Lee et al. (2018); Ren et al. (2018) have been proposed for predicting the lifetime distribution of a subject given her covariates, while effectively handling censored data that typically arise in survival tasks. DeepHit Lee et al. (2018) introduced a novel architecture and a ranking loss function in addition to the log-likelihood loss for lifetime prediction in the presence of multiple competing risks. Using a log-likelihood loss similar to DeepHit, Ren et al. (2018) propose a recurrent architecture to predict the survival distribution that captures sequential dependencies between neighbouring time points. To the best of our knowledge, there hasn't been a work on using neural networks for a lifetime clustering task.

**Frailty analysis.** Extensive research has been done on what is known as frailty analysis, for predicting survival outcomes in the presence of clustered observations (Hougaard, 1995; Chuang et al., 2005; Huang and Wolfe, 2002). Although frailty models provide more flexibility in the presence of clustered observations, they do not provide a mechanism for obtaining the clusters themselves, which is our primary goal.

