# OpenReview forum: "Deep Lifetime Clustering"
_ICLR.cc/2020/Conference — Reject_

### Official Review · AnonReviewer1 · 2019-10-21
**Official Blind Review #1**

**Rating:** 6

**Review:**

This paper proposes a deep learning method for clustering subjects (patients, users of social networks) into clusters of different lifetime (survival time) distributions. Inputs for a subject include covariates (age, gender, etc) and an event sequence. A key difficulty in the problem is that the lifetime of a subject is often unobserved. The proposed method is to learn a termination function that is non-decreasing with time, which essentially treats prolonged inactivity as termination in a probabilistic way. Clustering is done in a discriminative manner where the objective function considers cluster sizes besides between-cluster differences in lifetime distribution. (The inclusion of cluster size information in the objective function avoids degenerate clustering results.)
The paper is strong in technical contents. The objective function is well-motivated and placed on solid theoretical foundation. Empirically, the proposed method performed significantly better than baselines.
The weakness of the paper is that the utility of the work has not been demonstrated. For example, the Friendster dataset (1.1 million social network users) is partitioned into 2-5 clusters. However, are there clusters that a service provider can use to improve their service? It is doubtful whether such “useful” clusters can be found using the proposed algorithm. One might need to obtain a fairly large number of clusters, via conditioning perhaps, before finding some really useful ones.
•	The paper is strong in technical contents. The objective function is well-motivated and placed on solid theoretical foundation.
•	Empirically, the proposed method performed significantly better than baselines.
•	The utility of the work has not been demonstrated. Potential impact is an outstanding issue here because the proposed method is very special-purpose.

Question to the authors: It is observed that the numbers of friends and comments are strongly correlated with the clustering. Were those two covariates included in the analysis? If not, why?



**Experience Assessment:**

I have published one or two papers in this area.

**Review Assessment: Checking Correctness Of Derivations And Theory:**

I assessed the sensibility of the derivations and theory.

**Review Assessment: Checking Correctness Of Experiments:**

I assessed the sensibility of the experiments.

**Review Assessment: Thoroughness In Paper Reading:**

I read the paper at least twice and used my best judgement in assessing the paper.

---

> ### Author Response · Authors · 2019-11-09
> **Response to Review #1**
>
> Thank you for your positive comments and feedback.
>
> Q1. "Potential impact is an outstanding issue here because the proposed method is very special-purpose. Are there clusters that a service provider can use to improve their service? One might need to obtain a large number of clusters, via conditioning perhaps, before finding some really useful ones."
>
> Lifetime clustering is a very important task, with impactful applications in multiple fields.
>
> 1. There are numerous biomedical applications for this approach for example, clustering gene profiles to identify cancer subtypes [1,2,3], identifying subgroups of patients with different clinical characteristics [4].
> Concurrent to our work, Chen et al. [1] (October 2019) propose a supervised deep learning approach to identify cancer subtypes with different lifetime distributions. However, their approach requires at least a few ground truth lifetime clusters for supervision, whereas our approach does not require such supervision.
>
> 2. Similarly, identifying user subgroups in a social network can be a very useful tool. For example, a service provider might wish to obtain lifetime clusters of users based on their usage pattern alone (e.g., a high-risk cluster might only include people that use features X, Y and Z of the service). Identifying such groups can provide insights about the best/worst workflows in the service (different from predicting the lifetime of individual users). Note that the service provider need not obtain a large number of clusters to find these groups; she simply needs to remove all other covariates (like age, number of friends, etc.) whose analysis is not required.
>
> Moreover, of independent interest, the Kuiper loss provided by our paper (through our cheap-to-compute bound) is shown to be superior to MMD in our task. MMD finds a wide range of applications in machine learning. One could dedicate an entire paper just to the Kuiper bound; in our paper, it is just one of the contributions (albeit important).
>
> Q2. "Were number of friends and comments included in the analysis?"
> Yes, the number of friends and comments were used as covariates for lifetime clustering (along with 58 other covariates). Matching our intuition, number of friends and comments were correlated with the cluster assignments, thus qualitatively validating the clusters.
>
>
> References:
>
> [1] Chen, Runpu, et al. "Deep learning approach to identifying cancer subtypes using high-dimensional genomic data." Bioinformatics (2019).
> [2] Jiang, Limin, et al. "Discovering cancer subtypes via an accurate fusion strategy on multiple profile data." Frontiers in genetics 10 (2019): 20.
> [3] Guo, Yang, Xuequn Shang, and Zhanhuai Li. "Identification of cancer subtypes by integrating multiple types of transcriptomics data with deep learning in breast cancer." Neurocomputing 324 (2019): 20-30.
> [4] Hastie, Barbara A., et al. "Cluster analysis of multiple experimental pain modalities." Pain 116.3 (2005): 227-237.

---

### Official Review · AnonReviewer2 · 2019-10-24
**Official Blind Review #2**

**Rating:** 6

**Review:**

This paper proposes a method to cluster subjects based on the latent lifetime distribution. The proposed model clusters by maximizing the empirical divergence between different clusters.

The problem setting of this paper is not clear to me. I am not sure which variables are observed and which are not.  For example, in the Friendster experiment, the data of 5 months from joining is used for clustering. However, the termination windows are chosen to be 10 months. Therefore, it is clear that the observed data will not contain the termination signals, and I do not believe the training of the model is possible, without observing any termination signals.  In the paper, do we consider only one type or multiple types of events? Is $M_{k, i}$ a vector that represents the attributes or properties of an event?

Some details of the model are not clear to me. In Equation (2), the input of the neural network differs in length across different subject $u$, because the number of observed events for each subject is different.
How does the proposed neural network take inputs of different lengths?
How the non-decreasing function $\xi^{(u)}$ is defined in Section 3.2? Is it a function of the observed data for each subject?

How the empirical distribution $\hat{S}_i$ in Equation (4) is computed is also not clear to me. How $\hat{S}_i$ is a vector? Is it constructed by concatenating $\hat{S}_k(W_1, W_2; D)[t] $ with different $t$? How to normalize $\hat{S}_i$ such that it is a valid probabilistic distribution? Since $\hat{S}_i$ is high dimensional, it looks very challenging to estimate the joint distribution.

The overall objective function is given by Equation (4), is it correct? In Equation (4), why should we compute the minimum values across all possible pairs of clusters rather than the summation of all pairs? If Equation (4) is the overall objective function, then it looks like the model does not contain a component that maximizes a likelihood function. How is it guaranteed that the model will fit the data? It looks like the model will converge to a trivial solution that $\beta$ is a constant such that $\beta = 1$ for one cluster and $\beta = 0$ for another cluster, if the likelihood function is not involved. This will give a maximum divergence between distributions.

In summary, it seems that numerous technical details are missing and the paper might contain technical flaws. I do not suggest the acceptance of this paper.



**Experience Assessment:**

I do not know much about this area.

**Review Assessment: Checking Correctness Of Derivations And Theory:**

I carefully checked the derivations and theory.

**Review Assessment: Checking Correctness Of Experiments:**

I assessed the sensibility of the experiments.

**Review Assessment: Thoroughness In Paper Reading:**

I read the paper thoroughly.

---

> ### Author Response · Authors · 2019-11-09
> **Response to Review #2 (Part 1/2)**
>
> Thank you for the detailed review. We would like to clarify that the paper has absolutely no technical flaws, as we detail next.
>
> Q1. "The problem setting of this paper is not clear to me. I am not sure which variables are observed and which are not."
> To help the reader, we have moved the Table of notations from the Supplementary Material to the main paper (now Table 1).
> Since the clusters are unobserved, all variables with subscript $k$ are unobserved. In Definition 1, we use variables with subscript $k$ to indicate a random variable indexed by a random subject of cluster $k$. The definition describes only the underlying (hidden) generative process for the observed training data.
>
> Consider only the Friendster social network where $u$ denotes a user in the system. Then we consider the following variables:
> - $X^{(u)}$ (observed) is a vector of user covariates (e.g., age, gender).
> - $Y^{(u)}_i$ is the inter-event time between activity events (e.g., sending comments) and $M^{(u)}_i$ is a vector of event covariates.
>                 - For training users, the activity events are observed for a long time (within $[0, t_m]$) to compute observed lifetime $H^{(u)}$ and period of inactivity $\chi^{(u)}$ (both of these variables are tied to $t_m$).
>                 - For test users, the activity events are observed only till time $\tau = 5$ months from their joining. We wish to cluster the test users using their covariates and the initial activity events.
>  We have added a paragraph in the paper to clarify this distinction between training and test users.
>
> - $\theta^{(u)}$ (observed) is the joining time.
> - $A^{(u)}_i$ is the termination signal (observed in healthcare datasets, unobserved in social networks).
> - $T_k$ (unobserved) is the lifetime of a random subject in cluster $k$.
> - $S_k$ (unobserved) is the lifetime distribution (CCDF) $P[T_k > t]$ of cluster $k$.
>
>
> Q2. "In the paper, do we consider only one type or multiple types of events?"
> We consider different types of activity events (e.g., login, send/receive comments). Terminal event (e.g., quitting social media) is one that ends all activity events. Terminal event can have different causes; but lacking additional information about these causes, we do not model them separately in this paper.
>
> Q3. "Is $M_{k, i}$ a vector that represents the attributes or properties of an event?"
> Yes, $M_{k, i}$ is a random vector representing the attributes of the $i$-th activity event of a random user in cluster $k$. $M^{(u)}_i$ is the same for a particular user $u$.
>
> Q4. "In the Friendster experiment, the data of 5 months from joining is used for clustering ... termination windows are ... 10 months. Therefore, it is clear that the observed data will not contain the termination signals ... I do not believe the training of the model is possible without observing termination signals."
> Unobservability of termination signals is a property of the domain/dataset (healthcare or social networks) and not due to the choice of timeout window. For example, in Friendster, we do not know if or when the users deactivate their account; we only have information regarding their activities (i.e., that they have been inactive for say 15 months). In contrast, time of death (if occurred) is available in healthcare datasets.
>
> A "timeout" window of 10 months is an artificial way of providing these termination signals: if a user is inactive for greater than 10 months, then consider the user "terminated" (left the social network). Note that we use artificial termination signals only for baselines and to compute performance metrics, and not for training our model. We train our model by jointly learning a smooth "timeout" window in the form of $\beta^{(u)}$. Whereas we see only $\tau=5$ months of activities of test subjects, we observe training subjects for longer times (e.g., $[0, t_m]$ is 6 years long in Friendster experiments), thus facilitating model training.
>
> Q5. "How the non-decreasing function $\xi^{(u)}$ is defined in Section 3.2? Is it a function of the observed data for each subject?"
> As described in Section 3.2, we use $\beta^{(u)} := 1 - e^{-\xi^{(u)} \chi^{(u)}}$ with a shared scalar rate parameter $\xi^{(u)} = W_2$. This makes sure that $\beta^{(u)}$ is a non-decreasing function of inactivity; longer the inactivity, higher the probability of termination. To avoid confusion, we have replaced $\xi^{(u)}$ with $W_2$ in the paper.

---

> ### Author Response · Authors · 2019-11-09
> **Response to Review #2 (Part 2/2)**
>
> Q6. "How does the proposed neural network take inputs of different lengths?"
> As described in Section 3.1: "In our experiments, we compute summary statistics over the observed events $\{M^{(u)}_i, Y^{(u)}_i\}_i$ in order to make it compatible with the feedforward architecture." In Friendster experiments for example, we compute number of comments sent/received, number of people interacted with, mean inter-event times, etc. Note that our model is not restricted to a feedforward architecture, a recurrent architecture could be used instead. Our primary contribution is the loss function and a framework for lifetime clustering.
>
> Q7. "How the empirical distribution $\hat{S}_i$ in Equation (4) is computed is also not clear to me ... is a valid probabilistic distribution? Since $\hat{S}_i$ is high dimensional, it looks very challenging to estimate the joint distribution."
> $\hat{S}_k$ is simply a shorthand for the vector $\hat{S}_k(W_1, W_2; D)$; both denoting one-dimensional empirical lifetime distribution $P[T_k > t]$ of a cluster $k$. We use the notation $\hat{S}_k(W_1, W_2; D)$ to emphasize that it is a function of the model parameters $W_1$ and $W_2$; however, $\hat{S}_k$ is not a joint distribution of these parameters. We believe Table 1 (notations) will clarify this to the reader.
>
> Please see Figures (2b), (2c) and (3) for example lifetime distributions. Since we consider discrete times $t=0,1,2..$, $\hat{S}_k$ is a vector such that $\hat{S}_k[j] = P[T_k > j]$, i.e., the probability that the lifetime of a random subject in cluster $k$ is greater than $j$ time units.
>
> $\hat{S}_k$ is obtained using modified Kaplan-Meier estimates. As described in Section 3.3: "Kaplan-Meier estimates are a maximum likelihood estimate of the lifetime distribution of a set of subjects assuming (a) hard memberships and (b) the presence of termination signals. We modify the estimates to account for partial memberships and probability of termination instead". By construction (Proposition 1), $\hat{S}_k$ is a valid CCDF.
>
> Q8. "The overall objective function ... In Equation (4), why should we compute the minimum values across all possible pairs of clusters rather than the summation of all pairs?"
> Yes, the final objective function is given by Equation (4) which matches our lifetime clustering goal defined in Equation (1). We want *every* pair of clusters to have different survival distribution from each other. By maximizing the minimum divergence (say $\delta$), we make sure that every pair of clusters are at least $\delta$ apart. On the other hand, if we maximized the sum of divergences, then the model could maximize the divergence between one pair of clusters unboundedly while keeping all other of clusters very similar to one another; note that in this case, the sum of divergences will still be maximized, whereas the minimum divergence will not.
>
> Q9. "Equation (4) does not contain a component that maximizes a likelihood function ... It looks like the model will converge to a trivial solution that $\beta$ is a constant such ... divergence between distributions."
> The empirical lifetime distribution $\hat{S}_i$ of each of the clusters is obtained by maximizing the likelihood (please see Proposition 1 or Kaplan-Meier estimates). Additionally, $\beta^{(u)}$ are constrained by the non-decreasing function $\beta^{(u)} = 1 - e^{-W_2 \cdot \chi^{(u)}}$, where $\chi^{(u)}$ is the period of inactivity of user $u$. The non-decreasing function provides a structure such that the model cannot arbitrarily assign 0 and 1 to $\beta^{(u)}$. For example, if a user $u_1$ with 10 months of inactivity is assigned $\beta^{(u_1)} = 1$ then all users $u'$ with inactivity period greater than 10 months will be assigned $\beta^{(u')} = 1$ as well.
>
>
> We hope that we have addressed all of your concerns and convinced you that the paper does not contain technical flaws. Please let us know if you have additional concerns.

---

> > ### Comment · AnonReviewer2 · 2019-11-15
> > **Thank you for your response**
> >
> > I appreciate you added Table 1 in the revisions. It helps me understand the definition such as $\chi$ is the time elapsed and $S$ is a CCDF. Right now I am able to understand the model proposed in this paper. However, in table 1, I do not think $\widehat{S}$ is a "discrete vector" because each component in this vector has a continuous value between 0 and 1.
> >
> > The proposed method assumes that the membership of each subject is a function of the summary statistics over $M_i^{(u)}$ and $Y_i^{(u)}$; while all members in each cluster share the same life distribution. I suggest the authors state this assumption explicitly.
> >
> > I do not agree with the author that the minimum value in Equation (4) is a better choice than the summations. Consider that there are a pair of $\hat{S}_i$ and $\hat{S}_j$ are very similar to each other, while the differences between other pairs are moderate. If we choose to optimize for the minimum value in Equation (4),  the model will update $\hat{S}_i$ and $\hat{S}_j$ only; while all clusters other $i$ and $j$ are ignored in the optimization, despite that the divergences between some other pairs of clusters might still be increased. When we optimize for the summation, all pairs of clusters, rather than the most similar pair, will be involved simultaneously in the optimization. I believe that summation is a better choice for the objective function, especially when the number of clusters is large.
> >
> > I have updated my review rating because my concerns are addressed.

---

> > > ### Author Response · Authors · 2019-11-15
> > > **Thank you for updating your score**
> > >
> > >
> > > We have fixed the minor error in Table 1, thanks.
> > >
> > > We have stated the assumption about summary statistics in Section 3.1. Additionally, we have added sentences in both the Friendster and MIMIC experiments to remind the reader.
> > > We have added a sentence (last line of page 2) stating that all the users of a cluster share the same lifetime distributions (also stated in the Contributions paragraph).
> > >
> > > Q. I do not agree with the author that the minimum value in Equation (4) is a better choice than the summations. Consider that there are a pair of $\hat{S}_i$ and $\hat{S}_j$ are very similar to each other, while the differences between other pairs are moderate ... large.
> > > Yes, in the example provided, only $\Delta(\hat{S}_i$, $\hat{S}_j)$ will be optimized for the current iteration and possibly several subsequent iterations of the optimization. However, as the optimization progresses, $\Delta(\hat{S}_i^\text{new}$, $\hat{S}_j^\text{new})$ will become greater than the divergences between other pairs such that the minimum value in Equation (4) changes. That is, there will be $\hat{S}_k$ and $\hat{S}_l$ such that $\Delta(\hat{S}_k$, $\hat{S}_l) < \Delta(\hat{S}_i^\text{new}$, $\hat{S}_j^\text{new})$, and hence for that iteration, the model optimizes $\hat{S}_k$, $\hat{S}_l$. Continuing in this fashion, we can achieve higher divergence between every pair of clusters.
> > > We have performed experiments maximizing sum of divergences in the past, and have confirmed that it does not work (please see Figure 12 in the supplementary material). Here we provide an example against maximizing sum of divergences. Consider $K=4$ balanced clusters with lifetime distributions $\hat{S}_1 = \hat{S}_2 \neq \hat{S}_3 = \hat{S}_4$. Note that such a clustering is not desirable since there are virtually only 2 clusters found (as $\hat{S}_1$ and $\hat{S}_2$ are the same, etc). But for such an assignment, sum is relatively high $\sum_{i\neq j} \Delta(\hat{S}_i, \hat{S}_j) = 4 \cdot \Delta(\hat{S}_1, \hat{S}_3)$. The sum could be further optimized by simply increasing divergence between $\hat{S}_1$ and $\hat{S}_3$ while keeping all other clusters equal to one of $\hat{S}_1$ or $\hat{S}_3$.
> > > On the other hand, since $\min_{i\neq j} \Delta(\hat{S}_i, \hat{S}_j) = 0$, our model will maximize the divergence between $\hat{S}_1$ and $\hat{S}_2$ first.
> > >
> > > We have added section A.5 and Figure 12 in the supplementary material discussing the issues with using sum instead of minimum. We have also added a footnote below Definition 3 (clustering problem) pointing interested readers to section A.5.

---

### Official Review · AnonReviewer3 · 2019-10-26
**Official Blind Review #3**

**Rating:** 3

**Review:**

The authors propose an approach to cluster subjects into K clusters according to their underlying, often unknown, life distribution. The loss function to the neural-network-based approach is based on a tight upper bound to the two-sample Kuiper test. Experiments are conducted on artificial and two real life datasets.

I enjoyed reading the paper, the authors propose an approach to address an important yet relatively under-explored problem in survival analysis.

It is not entirely clear how to handle the case when after the model is trained, H^(u) for a new subject u is larger than t_max when the model is trained. In such case, \Chi^(u) will be negative, thus what happens with \beta^(u)(W_2)?

When the termination signals are not available (friendster data) termination signals are set artificially when training and evaluation, thus, is the model learning artificially set termination signals and thus artificial survival functions? what is the point of doing this? Moreover, if the termination signals are artificial, what is the point of calculating C-Index and Integrated Brier score? I understand the motivation to compare to existing methods, it is really not clear how meaningful they are at evaluating performance.

In general, using C-Index and Brier scores in the context of the presented application may be misleading (or non-applicable) because these metrics are usually applied to survival analysis scenarios where one seeks to estimate likelihood of survival or time to event (over a usually infinite time horizon). Here, \beta^(u)(W_2) is a function of \Chi^(u) which is known (also not comparable to survival analysis), artificially tied to a pre-specified time-horizon t_max, and not dependent on covariates as in survival analysis. Further, SSC-Bair and SSC-Gaynor are clustering models, how are survival estimates obtained for these?

In practice, how is K selected? based on the friendster results for K=2,3,4,5 they all seem to produce distinct clusters, so which one should be used? This raises a question: how are the clusters, their members or the number of clusters informing the use case? In the MIMIC III experiment, where events are observed, the model finds two clusters without strikingly different survival functions (relative to friendster). Can one really consider S_2 as the high-risk group? Can one get higher risk clusters with larger K?

Minor:
- A_i^(u) is missing from the definition in the training data.
- In Section 3.2, if \xi^(u) is shared, why the superscript indicating that is subject specific?

**Experience Assessment:**

I have published one or two papers in this area.

**Review Assessment: Checking Correctness Of Derivations And Theory:**

I assessed the sensibility of the derivations and theory.

**Review Assessment: Checking Correctness Of Experiments:**

I carefully checked the experiments.

**Review Assessment: Thoroughness In Paper Reading:**

I read the paper thoroughly.

---

> ### Author Response · Authors · 2019-11-09
> **Response to Review #3 (Part 1/2)**
>
> Thank you for your positive comments and feedback.
>
> Q1. "It is not entirely clear how to handle the case when after the model is trained, ... what happens with $\beta^{(u)}(W_2)$?"
> $\beta^{(u)}$ are only computed and used during training. For a new test subject, the model outputs cluster assignments $\alpha^{(u)}$ using only the covariates $X^{(u)}$ and the subject's events till time $\tau$ as described in Equation (2). We do not need to calculate $\beta^{(u)}$, $H^{(u)}$, $\chi^{(u)}$ for a test subject; indeed we will not be able to calculate them since they are all tied to $t_m$. $H^{(u)}$ and $\chi^{(u)}$ are only used during training to obtain a probabilistic proxy for true lifetime of $u$ via $\beta^{(u)}$.
> We have added a paragraph in Section 2 to clarify this distinction between training and test subjects.
>
> Q2. "When the termination signals are not available (friendster data) ... set artificially when training and evaluation, ... what is the point of doing this? ... what is the point of calculating C-Index and Integrated Brier score? I understand the motivation to compare to existing methods, it is really not clear how meaningful they are at evaluating performance."
> The paragraph titled "Termination signals for evaluation and baselines" (page 7 in the paper) clarifies this. The termination signals are artificially set only for the baselines (since they cannot handle unobservability), and during evaluation (since *all* the performance metrics including Logrank score require termination signals). Our approach is trained without any termination signals; they are learnt using the smooth "timeout" window $\beta^{(u)}$ as part of the optimization. This unfairly helps the competing methods since they are trained and evaluated using termination
> signals, whereas our approach is not trained with these termination signals.
>
> Q3. "In general, using C-Index and Brier scores ... may be misleading (or non-applicable) ... horizon. Here, $\beta^{(u)}(W_2)$ is a function of $\chi^{(u)}$ which is known ... and not dependent on covariates as in survival analysis. Further, SSC-Bair and SSC-Gaynor are clustering models, how are survival estimates obtained for these?"
> We used all metrics available in the literature, Logrank, C-index and Brier score to further validate the discriminative power of the clusters. This is a standard quantitative evaluation procedure used in unsupervised learning, common, for instance, in learning unsupervised image representations (e.g., [1]).
> The answer to (1) above should clarify that $\beta^{(u)}(W_2)$ and $\chi^{(u)}$ (both tied to $t_m$) cannot be computed for test subjects.
> The clusters from all the methods (SSC-Bair, SSC-Gaynor, DeepHit+GMM, DeepCLife) are evaluated the same way. Given the cluster assignments $\kappa(u') \in \{1, \ldots, K\}$ and the termination signals (possibly using a ``timeout'' window as described in answer to (2) above) for all the users $u'$ in the test data, we can obtain the empirical lifetime distribution of all the clusters $\hat{S}_k, \: \forall k \in \{1,\ldots,K\}$ using the Kaplan-Meier estimates (over the test data alone). Then, the empirical lifetime distribution of a user $u'$ is given by that of her assigned cluster, i.e., $\hat{S}^{(u')} := \hat{S}_{\kappa(u')}$ (shared for all users in the cluster). Integrated Brier score can be computed as usual using $\hat{S}^{(u')}$ (Equation 14). C-index is computed by using the expected lifetime obtained from the lifetime distribution $\hat{S}^{(u')}$ as the predicted lifetime of $u'$.
> We have updated the Metrics subsection in the paper to include the above discussion.
>
> References:
> [1] Bengio, Yoshua. "Deep learning of representations for unsupervised and transfer learning." Proceedings of ICML workshop on unsupervised and transfer learning. 2012.

---

> ### Author Response · Authors · 2019-11-09
> **Response to Review #3 (Part 2/2)**
>
>
> Q4. "In practice, how is K selected? based on the Friendster results for K=2,3,4,5 they all seem to produce distinct clusters, so which one should be used?"
> Similar to standard clustering algorithms, the choice of $K$ is left to the practitioner; however, the validation performance of the clusters with respect to the Logrank scores can act as a useful guide. We have added section A.4 (pg 17-18) and Figures 10 and 11 in the supplementary material discussing this heuristic to select the optimal number of clusters.
> Figure 10 shows the Logrank scores on a held-out validation set for clusters obtained from DeepCLife-KuiperUB on the Friendster dataset with $K=2\ldots 7$. We observe that $K=4$ gives the best mean Logrank score and $K=5$ is close behind. However, for $K>5$, the Logrank scores drop drastically indicating that there are no more clusters of interest with distinct lifetime distributions from the ones already found.
> Please see Figure 11 for a similar discussion on MIMIC-III results based on Logrank scores for $K=2\ldots 8$.
>
> Q5. "This raises a question: how are the clusters, ... informing the use case? In the MIMIC III experiment, ... finds two clusters without strikingly different survival functions (relative to Friendster). Can one really consider $S_2$ as the high-risk group? Can one get higher risk clusters with larger K?"
> The clusters found by DeepCLife-KuiperUB in MIMIC-III dataset with $K=2$ are significantly different as evaluated by the Logrank test (score $=205.25$, p-value $\approx$ 0). Visual comparison of the clusters' lifetime ditributions across datasets can be misleading, especially since the timescale is very different (50 months in Friendster vs 30 days in MIMIC-III). In prior works (e.g., [2]), typically any two clusters with Logrank p-value less than 0.05 have been classified as high-risk/low-risk.
> We have added Figure 9 in the supplementary material showing the empirical lifetime distributions of the clusters found by DeepCLife-KuiperUB in the MIMIC-III dataset for $K=2...6$. We observe that there are no considerably higher risk clusters for larger values of $K$ than the one already obtained with $K=2$. However, we do obtain a very-low-risk cluster for $K=3,4,6$.
> We have provided a brief qualitative analysis of the clusters found by DeepCLife-KuiperUB in the Friendster dataset (please see last paragraph of Friendster experiment). We show that the clusters align with our intuition, i.e., a subject in the high-risk cluster has on average far fewer friends (1.56 vs 7.76) and sends fewer comments (1.07 vs 5.06) in the initial $\tau=5$ months.
>
> Q6. "Minor: $A_i^{(u)}$ is missing from the definition in the training data. In Section 3.2, if $\xi^{(u)}$ is shared, why the superscript indicating that is subject specific?"
> We describe the termination signals in the same paragraph describing the training data, but separate from other variables because the availability of termination signals depends on the domain (from the paper: "The training data may or may not contain the termination signals, $A^{(u)}_i$ ... ").
> We have replaced $\xi^{(u)}$ with $W_2$ to avoid confusion.
>
> References:
> [2] Bair, Eric, and Robert Tibshirani. "Semi-supervised methods to predict patient survival from gene expression data." PLoS biology 2.4 (2004): e108.

---

### Decision · Program_Chairs · 2019-12-19

**Decision:**

Reject

**Comment:**

The authors propose a clustering algorithm for users in a system based on their lifetime distribution. The reviewers acknowledge the novelty of the proposed clustering algorithm, but one concern left unresolved is how the results of the analysis can be of use in the real world examples used.